# Oligomerization-dependent and synergistic regulation of Cdc42 GTPase cycling by a GEF and a GAP

Sophie Tschirpke [ID], Werner K-G Daalman, Frank van Opstal & Liedewij Laan [ID] ✉

## Abstract

**Cell polarity is a crucial biological process essential for cell division, directed growth, and motility. In *Saccharomyces cerevisiae*, polarity establishment centers around the small Rho-type GTPase Cdc42, which cycles between GTP-bound and GDP-bound states, regulated by GEFs like Cdc24 and GAPs such as Rga2. To dissect the dynamic regulation of Cdc42, we employed in vitro GTPase assays, revealing inverse concentration-dependent profiles for Cdc24 and Rga2: with increasing concentration, Cdc24's GEF activity is nonlinear and oligomerization-dependent, which is possibly linked to the relief of its self-inhibition. In contrast, Rga2's GAP activity saturates, likely due to self-inhibition upon oligomerization. Together, Cdc24 and Rga2 exhibit a strong synergy driven by weak Cdc24–Rga2 binding. We propose that the synergy stems from Cdc24 alleviating the self-inhibition of oligomeric Rga2. We believe this synergy contributes to efficient regulation of Cdc42's GTPase cycle over a wide range of cycling rates, enabling cells to resourcefully establish polarity. As Cdc42 is highly conserved among eukaryotes, we propose the GEF–GAP synergy to be a general regulatory property in other eukaryotes.**

**Keywords** GTPase Activity; Cdc42 Regulation; GEF-GAP Synergy; Enzyme Kinetics; Oligomerization
**Subject Category** Cell Adhesion, Polarity & Cytoskeleton

## Introduction

Cells require robust, yet adaptable, functioning to survive in an ever-changing environment. One of such functionalities is cell polarity, which is essential for processes such as cell division, directed growth and secretion, and motility (Vendel et al, 2019). A well-studied system for polarity establishment is that of *Saccharomyces cerevisiae*: here the cell division control protein Cdc42 exits the cytoplasm and accumulates in one spot on the cell membrane, marking the site of bud emergence (Fig. 1A) (Nelson, 2003; Etienne-Manneville, 2004; Thompson, 2013; Diepeveen et al, 2018;

Chiou et al, 2017; Martin, 2015). Cdc42 is a highly conserved small Rho-type GTPase, whose activity involves cycling between a GTP-bound state and a GDP-bound state, a process tightly controlled by guanine nucleotide exchange factors (GEFs), such as Cdc24 and Bud3, and GTPase-activating proteins (GAPs), such as Rga1, Rga2, Bem2, and Bem3 (Park and Bi, 2007; Martin, 2015) (Fig. 1B).

A thorough understanding of Cdc42 GTPase activity and its regulation is crucial, as Cdc42 GTPase cycling is required for polarity establishment, and cells in which it has been impaired fail to polarize (Wedlich-Soldner et al, 2004). Cdc42 GTPase activity and its role in polarity establishment have been studied both in vivo (Smith et al, 2013; Lee et al, 2015; Sartorel et al, 2018) and in vitro. However, interpreting in vivo studies can be challenging due to the complexity within the regulatory network surrounding Cdc42. For instance, several proteins belong to the same group of regulators, leading to redundancy within the network. Further, many polarity proteins also interact with a variety of other polarity proteins, resulting in a dynamic assembly of loosely interacting proteins at the polarity spot (Gao et al, 2011; Daalman et al, 2020) (Fig. 1C). In vitro studies, where interactions can be probed in a highly controlled environment, are still comparatively scarce, focus on a small aspect of the system, and did not include interactions between different full-length regulatory proteins (Zheng et al, 1993; Zheng et al, 1994, Zheng et al, 1995; Zhang et al, 1997; Zhang and Zheng, 1998; Zhang et al, 1999; Zhang et al, 2001; Smith et al, 2002; Das et al, 2012; Johnson et al, 2009; Johnson et al, 2012; Golding et al, 2019). Therefore, they neglect effects present in multi-protein systems as occur in the dynamic and interconnected structure of the polarity network in vivo (Fig. 1C). A recent study confirms this notion, revealing that the scaffold Bem1 enhances the GEF activity of Cdc24 (Rapali et al, 2017).

To enhance our knowledge of Cdc42 regulation in *S. cerevisiae*, we employ a bulk in vitro GTPase assay to study properties of the entire Cdc42 GTPase cycle and its regulation by the GEF Cdc24 and GAP Rga2. These proteins are interesting study targets; Cdc24 is an essential protein and one of the two GEFs present at the polarity site (Daalman et al, 2020), and Rga2 has not yet been characterized as a full-length protein (Smith et al, 2002). Both proteins have the potential to oligomerize (Mionnet et al, 2008; Tarassov et al, 2008; Schlecht et al, 2012), a property which may enhance polarity establishment (Lang and Munro, 2022) and could

Bionanoscience Department, Delft University of Technology, Delft, The Netherlands. ✉E-mail: L.Laan@tudelft.nl

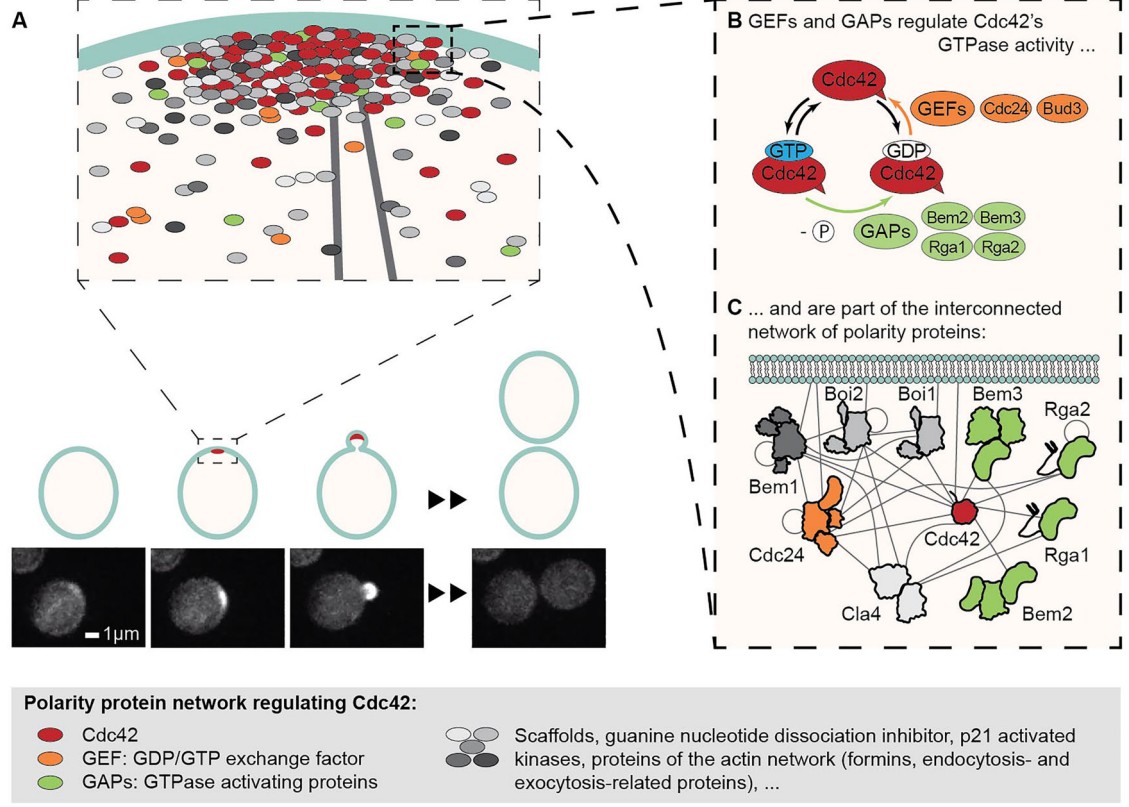

**Polarity protein network regulating Cdc42:**

- ● Cdc42
- ● GEF: GDP/GTP exchange factor
- ● GAPs: GTPase activating proteins

Scaffolds, guanine nucleotide dissociation inhibitor, p21 activated kinases, proteins of the actin network (formins, endocytosis- and exocytosis-related proteins), ...

**Figure 1. A complex network of polarity proteins regulates Cdc42 activity in vivo to establish cell polarity.**

(A) The accumulation of the small Rho-GTPase Cdc42 in one spot on the cell membrane establishes cellular polarity, initiating cell division of *Saccharomyces cerevisiae*. Polarity establishment is driven by interactions between Cdc42 and polarity proteins of an intricate network. Cdc42 is shown in red in the cartoon and in white in live cell microscopy images (bottom). (B) The polarity protein groups of GEFs and GAPs regulate Cdc42 GTPase cycling, a process required for its functioning in polarity establishment. (C) GEFs and GAPs are part of the interconnected and complex polarity protein network. This cartoon shows a subset of polarity proteins, and gray lines depict experimentally determined (direct or indirect) interactions. Protein subdomains roughly correspond to domain size but do not accurately depict the domain structure.

tune their activity in a dosage-dependent fashion. Lastly, Cdc24 and Rga2 regulate distinct steps of the Cdc42 GTPase cycle and interact in vivo (although it is unclear whether their interaction is direct or indirect) (McCusker et al, 2007; Breitkreutz et al, 2010; Chollet et al, 2020). In combination, Cdc24 and Rga2 could synergize, inhibit each other, or have no interplay. Such a GEF–GAP interaction is theorized to play a significant role in G-protein signaling (Ross, 2008).

Our data reveal distinct concentration- and oligomerization-dependent profiles for the GEF and GAP: specifically, Cdc24 stimulates Cdc42 GTPase activity in a nonlinear fashion, a process that is linked to its oligomerization and may relieve its self-inhibition. Rga2 also forms oligomeric structures, driven by its C-terminal GAP domain. Interestingly, the GAP activity of full-length Rga2, unlike that of its isolated GAP domain, saturates at an approximate 1:1 ratio of Rga2 to Cdc42. This saturation may be linked to self-inhibition upon oligomerization. In combination, Cdc24 and Rga2 exhibit a protein-specific GEF–GAP synergy, which appears to arise from weak Cdc24–Rga2 binding. We speculate that it arises from the relief of oligomeric Rga2's self-inhibition through Cdc24.

# Results

## Cdc42 GTPase activity can be reconstituted in vitro

We used the GTPase-Glo™ assay (Promega) to examine the entire GTPase cycle of Cdc42. This allowed us to study the interplay of regulators acting on different steps of the GTPase cycle. In each assay, Cdc42, alone or in combination with one or multiple effector proteins, is mixed with GTP and incubated at 30 °C for 1–1.5 h for GTPase cycling to occur. Then the reaction is stopped by transforming the remaining GTP to luminescent ATP. The luminescent signal of each reaction mixture is measured (Fig. 2). The decrease of the GTP concentration during GTPase cycling is well fitted by an exponential model (Appendix Supplementary Text 1, Appendix Fig. S1a–c):

$$[\text{GTP}]_t = [\text{GTP}]_{0h} \exp(-Kt)$$
$$[GTP]_{0h} = 1$$
$$K = K_1 + K_2 + K_{3,X1} + K_{3,X2} + K_{3,X1,X2}$$
$$= k_1[\text{Cdc42}] + k_2[\text{Cdc42}]^2 + k_{3,X1}[\text{Cdc42}][\text{X1}]^n + k_{3,X2}[\text{Cdc42}][\text{X2}]^m$$
$$+ k_{3,X1,X2}[\text{Cdc42}][\text{X1}]^n[\text{X2}]^m$$

(1)

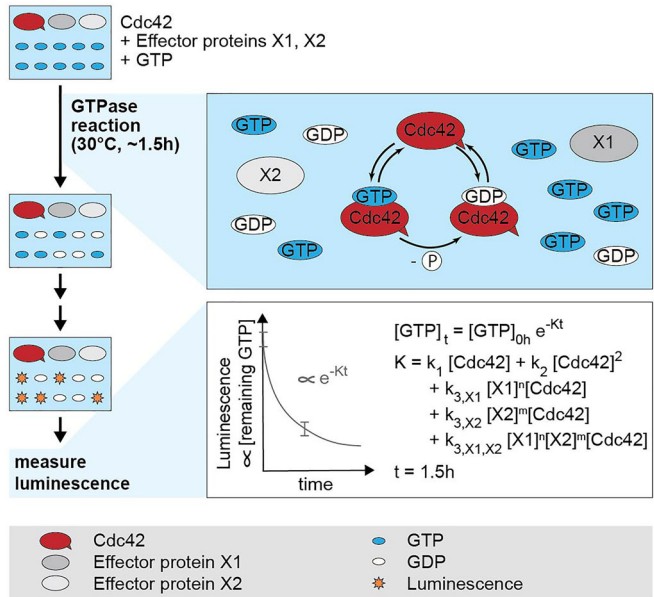

**Figure 2. Reconstitution of Cdc42 GTPase activity regulation in vitro.**

Bulk in vitro GTPase assays (GTPase-Glo™ assay, Promega) allow the study of the GTPase cycling of Cdc42. Cdc42, alone or in combination with effector proteins, is incubated with GTP at 30 °C for ~1.5 h, during which GTPase cycling occurs. After two processing steps, the amount of remaining GTP, measured as luminescence, is assessed. GTP hydrolysis cycling rates can be extracted by fitting the data with an exponential model (Appendix Supplementary Text 1) (Tschirpke et al, 2024).

where $K$ refers for the overall GTP hydrolysis rate, $X1$ and $X2$ are effector proteins, such as GEFs and GAPs, and $n$ and $m$ are natural numbers. From these fits, the GTPase cycling rates $k$ are determined (Figs. 3D, 4D, and 5D) (Tschirpke et al, 2024).

We first analyzed the GTPase activity of Cdc42 alone (Fig. 3A). To describe Cdc42 GTPase cycling, we included terms that depend linearly ($K_1$) and quadratically ($K_2$) on the Cdc42 concentration, the latter representing any possible effects due to cooperativity from dimeric Cdc42. While in vivo data suggest that *S. cerevisiae* Cdc42 does not dimerize (Kang et al, 2010), our in vitro GTPase assays revealed variability in Cdc42 behavior, with some Cdc42 constructs exhibiting quadratic rate increases and others showing linear ones (preprint: Tschirpke et al, 2023b). As our goal was to explore Cdc42–effector interactions, we used the general phenomenological description shown in Eq. (1) to fit the Cdc42 data (Fig. 3; Table 1). We remain cautious about deriving conclusions from this data on Cdc42 dimerization.

## The GEF activity of Cdc24 increases upon oligomerization

We next investigated how Cdc24 affects Cdc42 GTPase activity. Cdc24 is a known GEF, increasing the speed of the GDP release step. In agreement with previous studies (Rapali et al, 2017), sub-μM concentrations of Cdc24 substantially boost Cdc42's GTPase activity; the rate-contribution of Cdc24 is two orders of magnitude greater than those of Cdc42 alone (Fig. 3E; Table 2). We find that Cdc24's effect increases non-linearly with its concentration

(Fig. 3B), which could be explained by Cdc24 di- or oligomerization: Cdc24 has the capability to oligomerize via its DH domain (Mionnet et al, 2008). We attempted to reproduce Cdc24 oligomerization using size-exclusion chromatography–multi-angle light scattering (SEC–MALS) experiments, but did not observe Cdc24 oligomers (Appendix Fig. S3). Given that Cdc24 oligomerizes via weak interactions (Mionnet et al, 2008) and that SEC–MALS experiments are performed under a constant flow that could pull weakly bound Cdc24 oligomers apart, our results do not contradict previous findings. Dimers and oligomers could have an increased GEF activity through releasing Cdc24 from its self-inhibited state (Shimada et al, 2004). With increasing Cdc24 concentration, the amount of Cdc24 oligomers increases, resulting in a nonlinear increase of the overall GTP hydrolysis rate $K$.

To investigate our hypothesis, we utilized Cdc24 mutants Cdc24-DH3 and Cdc24-DH5, which exhibit 2.5× and 10× reduction in their oligomerization capacity (Mionnet et al, 2008). If the nonlinear rate increase by Cdc24 (and thus its GEF activity) is linked to its oligomerization, we expect the oligomerization-reduced mutants to (1) show a reduced GEF activity and (2) boost the overall GTP hydrolysis rate $K$ in a more linear fashion than Cdc24. A detailed discussion of our analysis is given in Appendix Supplementary Text 2. In brief, Cdc24-DH5 did not express as a full-length protein (Appendix Fig. S4). Cdc24-DH3 exhibited a 17× reduced GEF activity (Fig. 3C,E; Table 2) and increased the overall GTPase cycling rate $K$ in a 1.7× more linear fashion than wild-type Cdc24 (Appendix Table S1), confirming our hypothesis. Further, our data shows that previous findings on the in vitro GEF activity of peptides containing Cdc24 fragments do not translate to full-length Cdc24: Mionnet et al found that a chemically induced oligomerization of peptides based on Cdc24 fragments does not affect their GEF activity, suggesting that the GEF activity might be oligomerization-independent (Mionnet et al, 2008). We found that Cdc24-DH3, a Cdc24 mutant with reduced oligomerization capacity, has reduced GEF activity, questioning the generalizability of data based on protein fragments.

In summary, the data show that Cdc24 is active at sub-μM concentrations, boosting the overall GTP hydrolysis rate in a nonlinear fashion. We believe the non-linearity is linked to Cdc24 oligomerization and the subsequent release of Cdc24's self-inhibition upon oligomerization.

## The GAP Rga2 self-inhibits upon oligomerization

Next to GEFs, GAPs also increase Cdc42 GTPase cycling, but by catalyzing the GTP hydrolysis step. We investigated the effect of the GAP Rga2, which so far has not yet been characterized as a full-length protein. Rga2 has been shown to self-interact in vivo (Tarassov et al, 2008; Schlecht et al, 2012), but it is unclear if this interaction is direct or indirect (i.e., mediated via other proteins). Similar to Cdc24, Rga2 dimers or oligomers could affect its catalytic activity.

We first investigated Rga2 oligomerization using size-exclusion chromatography–multi-angle light scattering (SEC–MALS) experiments (Fig. 4A). Here, a protein sample is separated by size under flow using size-exclusion chromatography, where larger protein complexes elute first (at smaller volumes). The molecular weight of each fraction is then determined using multi-angle light scattering. We performed experiments using His-Rga2 (Rga2 tagged with an N-terminal 6His purification tag, in the following referred to as

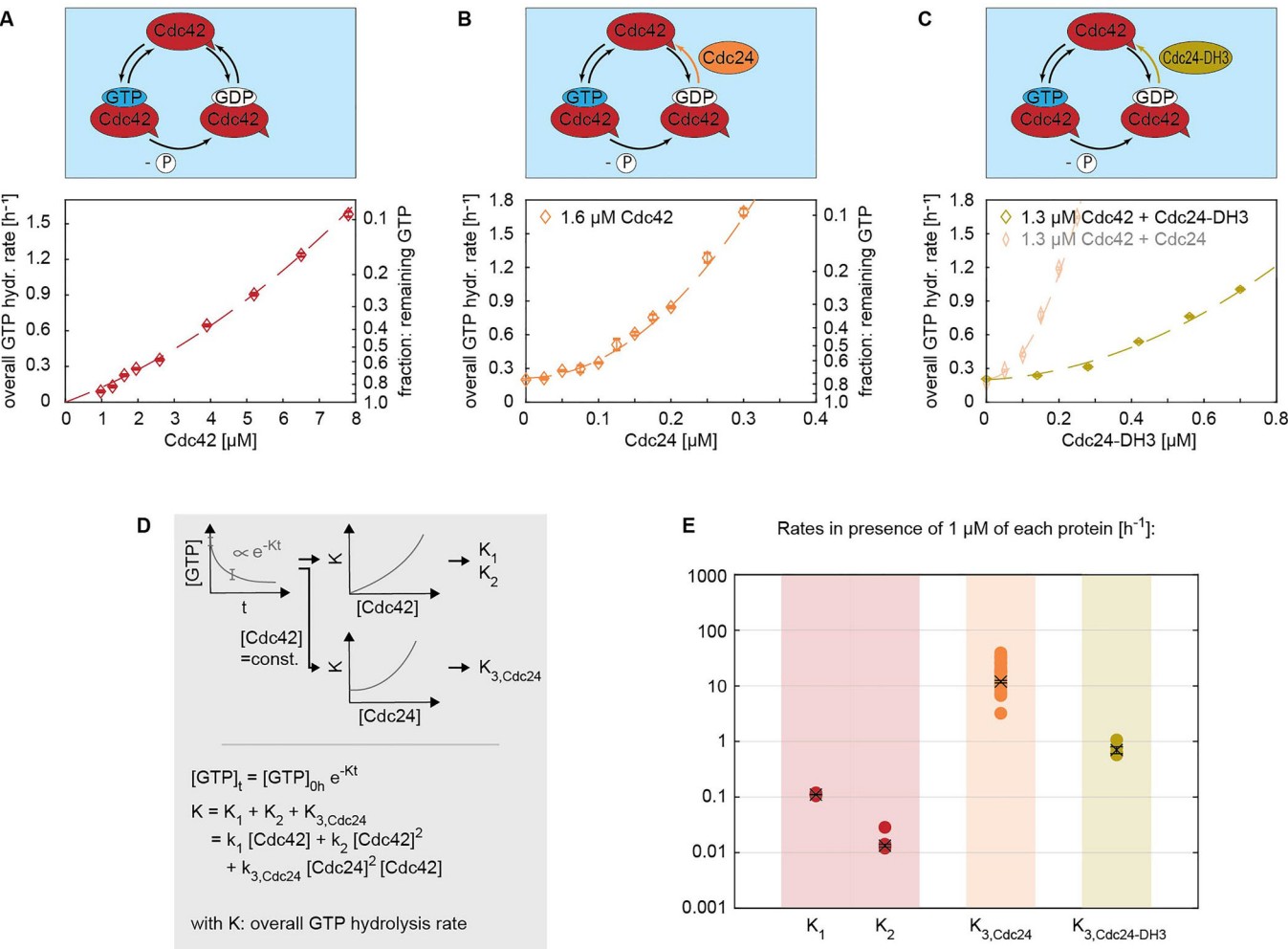

Figure 3. The strong GEF activity of Cdc24 requires oligomerization.

(A) The overall GTP hydrolysis rate $K$ of the GTPase Cdc42 in the absence of any effector proteins (graph shows data of a single measurement ($n = 1$)). (B) Cdc24 boosts the GTPase activity of Cdc42 in a nonlinear fashion and is active even at sub-µM concentrations (graph shows data of a single measurement ($n = 1$)). (C) Cdc24-DH3, a Cdc24 mutant with reduced oligomerization capacity, exhibits a weaker GEF activity (graph shows data of a single measurement each ($n = 1$)). (D) Illustration of the data processing and fitting model. (E) Summary of the rates $K$. The values shown refer to the rate values in the presence of 1 µM of each protein, e.g., "$K_{3,Cdc24}$" refers to "$k_{3,Cdc24}[Cdc24]^2[Cdc42]$" with $[Cdc42]=[Cdc24]=1\,\mu M$. Crosses with error bars represent the weighted mean and the standard error of the mean (Appendix Supplementary Text 1), filled circles show individual measurements. The numbers and number of measurements ($n$) are given in Table 2. Source data are available online for this figure.

Rga2$_{(I)}$) and a peptide containing only the GAP domain of Rga2 (Smith et al, 2002). In these in vitro experiments (Fig. 4A), Rga2$_{(I)}$ was only detected as higher-order oligomers, with no monomers present. The GAP domain appeared as four closely overlapping peaks, suggesting the existence of higher than monomeric structures. Because the peaks overlapped so closely, the exact masses and thus oligomeric states could not be determined. The SEC–MALS data show that Rga2 self-interacts directly, forming oligomers. Further, it implies that Rga2 oligomerization is facilitated by its C-terminal GAP domain.

Next, we assessed the GAP activity of Rga2 through GTPase assays, examining Rga2's concentration-dependent behavior. Rga2$_{(I)}$ increases the overall GTP hydrolysis cycling rate $K$ of Cdc42 in a linear fashion up until ~1.2 µM (corresponding to a 1:1 Rga2:Cdc42 ratio), after which its effect saturates (Fig. 4B). We also assessed a second Rga2

construct (His-Rga2-Flag, in the following referred to as Rga2$_{(II)}$), which showed exactly the same behavior (Fig. 4B). One possible explanation is that at ~1.2 µM Rga2 the GTP hydrolysis step becomes rate-limiting; at this Rga2 concentration the hydrolysis reaches its maximum rate and cannot be further accelerated. Alternatively, at higher concentrations, Rga2 may self-inhibit. To investigate this, we analyzed the activity of Rga2's GAP domain (Fig. 4C). The GAP domain displayed a similar linear rate increase, with rates approximately matching those of Rga2$_{(I)}$ and Rga2$_{(II)}$ (Fig. 4E; Table 2), yet without saturation, showing a linear increase across the full concentration range tested (up to 5 µM). This shows that the GTP hydrolysis step does not become rate-limiting at ~1 µM Rga2, supporting the possibility of Rga2 self-inhibition.

We speculate that the self-inhibition may originate from Rga2 oligomerization. Taken together, our data show that Rga2 has

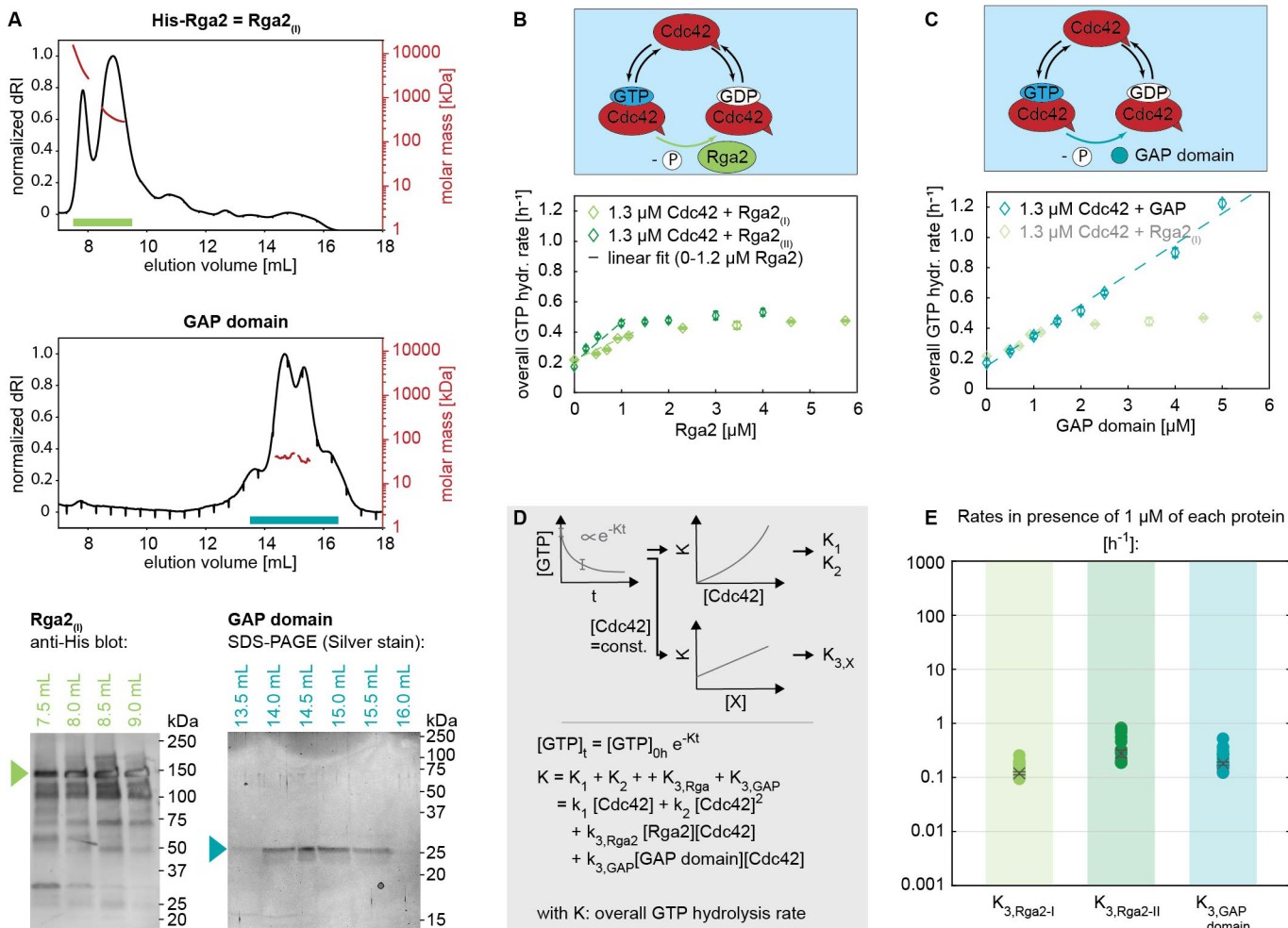

**Figure 4. The saturation of the GAP activity of Rga2 may be linked to Rga2 oligomerization.**

(A) Size-exclusion profile and MALS analysis (top) and SDS–PAGE/western blot analysis of SEC–MALS elution fractions (bottom) of His-Rga2 (Rga2$_{(I)}$) and Rga2's GAP domain: Rga2 self-interacts, forming higher-order oligomeric structures. Rga2 oligomerization is facilitated by its C-terminal GAP domain, which in SEC–MALS elutes in multiple states (mixtures of monomers and higher than monomer structures). (B) Rga2$_{(I)}$ and Rga2$_{(II)}$ increase Cdc42 GTPase cycling in a linear fashion up until ~1.2 µM, after which their effect saturates (graphs show data of a single measurement each ($n = 1$)). (C) Cdc42 GTPase cycling is accelerated by the GAP domain at a rate akin to Rga2, yet without its effect saturating (graphs show data of a single measurement each ($n = 1$)). (D) Illustration of the data processing and fitting model. (E) Summary of the rates $K$: Rga2$_{(I)}$, Rga2$_{(II)}$, and the GAP domain have a similar rate. The values shown refer to the rate values in the presence of 1 µM of each protein, e.g., "$K_{3,Rga2}$" refers to "$k_{3,Rga2}$[Rga2][Cdc42]" with [Cdc42] = [Rga2] = 1 µM. Crosses with error bars represent the weighted mean and the standard error of the mean (Appendix Supplementary Text 1), filled circles show individual measurements. The numbers and number of measurements ($n$) are given in Table 2. Source data are available online for this figure.

unique features distinct from the GAP domain, emphasizing that its full functionality is not confined to the GAP domain alone: (1) Rga2 self-interacts and forms oligomeric structures, facilitated by its C-terminal GAP domain. (2) The GAP activity of Rga2, unlike the activity of the GAP domain alone, saturates in GTPase assays. This saturation may be caused by self-inhibition upon oligomerization.

## Cdc24 and Rga2 synergistically regulate Cdc42 GTPase cycling

After having analyzed the individual effects of the GEF Cdc24 and the GAP Rga2, we explored their joint impact on Cdc42 GTPase cycling. This is noteworthy because (1) Cdc24 and Rga2 interact

in vivo (directly or indirectly) (McCusker et al, 2007; Breitkreutz et al, 2010; Chollet et al, 2020) and (2) the interplay between GEFs and GAPs is theorized to play a significant role in G-protein signaling (Ross, 2008).

We conducted GTPase assays containing Cdc42–Cdc24–Rga2$_{(I)}$ mixtures and fitted the data (Eq. (1); Fig. 5E) to obtain rates $K$.

For such three-protein mixtures, we obtain $K_{3,Cdc24}$ and $K_{3,Rga2}$, which represent the individual rate contributions of the GEF and GAP, and the interaction term $K_{3,Cdc24,Rga2}$. A positive interaction term represents synergy between the GEF and GAP, a negative value represents inhibition. If the term is zero, neither protein would affect the other.

In presence of both Cdc24 and Rga2$_{(I)}$, the GTPase activity of Cdc42 increases drastically (Fig. 5A). The contributions of the

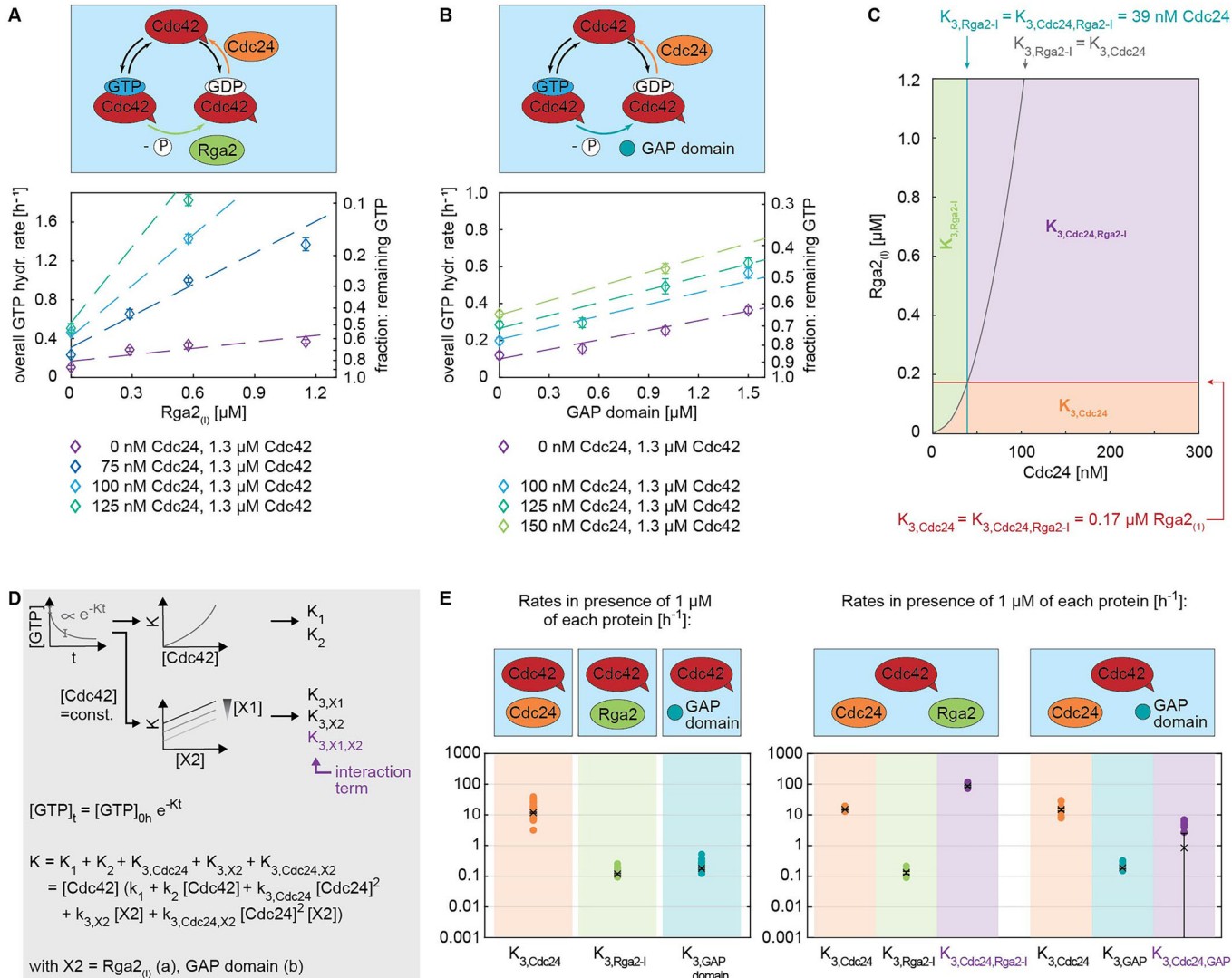

**Figure 5. The GEF Cdc24 and GAP Rga2 regulate Cdc42 GTPase cycling synergistically.**

(A, B) GTPase assay data of (A) Cdc42–Cdc24–Rga2$_{(I)}$ and (B) Cdc42–Cdc24–GAP domain mixtures (the graphs show data of a single measurement each ($n = 1$)). (C) Regime diagram illustrating the dominant rate $K$ across varying Cdc24 and Rga2$_{(I)}$ concentrations; above 39 nM Cdc24 and 0.17 μM Rga2, the interaction term $K_{3,Cdc24,Rga2}$ becomes predominant. The diagram was generated using rates $k_3$ for effectors Cdc24 and Rga2$_{(I)}$ given in Table 3. An extended diagram is shown in Appendix Fig. S10. (D) Illustration of the data processing and fitting model. (E) Summary of the rates obtained in the three-protein assay (right) in comparison to those of the two-protein assay (left): In three-protein assays, the rate contribution of the individual proteins is comparable to those obtained in the two-protein assay. In addition, an interaction rate is obtained (shown in purple). This rate is large in the case of Cdc42–Cdc24–Rga2$_{(I)}$ mixtures ($K_{3,Cdc24,Rga2} \approx 87$), indicating a synergy between Cdc24 and Rga2$_{(I)}$. For Cdc42–Cdc24–GAP domain mixtures, the interaction rate is zero ($K_{3,Cdc24,GAP\ domain} \approx 0$), revealing no synergy. The values shown refer to the rate values in the presence of 1 μM of each protein, e.g., "$K_{3,Cdc24}$" refers to "$k_{3,Cdc24}$ [Cdc24]$^2$ [Cdc42]" with [Cdc42]=[Cdc24]=1 μM. Crosses with error bars represent the weighted mean, and the standard error of the mean (Appendix Supplementary Text 1), and filled circles show individual measurements. The numbers and number of measurements ($n$) are given in Table 3. Source data are available online for this figure.

individual proteins ($K_{3,Cdc24}$, $K_{3,Rga2}$) are roughly the same as in assays containing only one of the effectors (Fig. 5D; Table 3). Importantly, the interaction term $K_{3,Cdc24,Rga2}$ is the dominating term for essentially the entire concentration regime (i.e., above 0.17 μM Rga2$_{(I)}$ and 39 nM Cdc24, Fig. 5C; Appendix Fig. S10), suggesting a GEF–GAP synergy.

To ensure that a positive $K_{3,Cdc24,Rga2}$ is not an assay artifact, we performed additional GTPase assays using proteins considered inert (bovine serum albumin (BSA), Casein), as well as mixtures of BSA with Cdc24 and BSA with Rga2$_{(I)}$ (Appendix Supplementary

Text 3, Appendix Figs. S5a,b, S6a–d, S7a–c, S8a–d, Appendix Tables S2, S3, and S4). In these assays, no significant synergy occurred, confirming that $K_{3,Cdc24,Rga2}$ is a genuine signal.

The positive $K_{3,Cdc24,Rga2}$ could originate from two sources: (1) Cdc24–Rga2 synergy due to rate-limiting effects of different steps in the GTPase cycle, and (2) Cdc24–Rga2 synergy due to protein-protein interactions.

Could rate-limiting steps be the *only* source of the positive $K_{3,Cdc24,Rga2}$? The rate-limiting step model is a common framework used to describe the cycling behavior of GTPases. It assumes that at least

**Table 1. GTP hydrolysis cycling rates $k_1$ and $k_2$ of Cdc42.**

|  | $k_1$ [µM$^{-1}$ h$^{-1}$] | $k_1$ std. err. [µM$^{-1}$ h$^{-1}$] | $k_2$ [µM$^{-2}$ h$^{-1}$] | $k_2$ std. err. [µM$^{-2}$ h$^{-1}$] |
|---|---|---|---|---|
| Pooled estimate ($n = 5$) | 0.110 | 0.002 | 0.013 | 0.001 |

**Table 2. Interaction rates $k_{3,X}$ of Cdc42 - effector protein mixtures.**

|  | Effector protein X | $k_{3,X}$ [µM$^{-2}$ h$^{-1}$]* | $k_{3,X}$ std. err. |
|---|---|---|---|
| Pooled est. ($n = 34$) | Cdc24 | 11.887 | 0.594 |
| Pooled est. ($n = 4$) | Cdc24-DH3 | 0.702 | 0.100 |
| Pooled est. ($n = 17$) | Rga2$_{(I)}$ | 0.119 | 0.008 |
| Pooled est. ($n = 12$) | Rga2$_{(II)}$ | 0.278 | 0.042 |
| Pooled est. ($n = 20$) | GAP domain | 0.181 | 0.013 |

*Unit in case of X=Cdc24: [µM$^{-3}$ h$^{-1}$].

one of the three GTPase cycle steps—GTP binding, GTP hydrolysis, or GDP release—is rate-limiting. It has the potential to explain the synergy term we observed in our data: if both the GDP release step, accelerated by the GEF Cdc24, and the GTP hydrolysis step, accelerated by the GAP Rga2, are rate-limiting, then adding both effectors would accelerate the overall cycle more than either alone. This synergistic increase in cycling speed would result from their combined effect on the GTPase cycle and not from direct interaction between the proteins. The effect size of the synergetic term, as predicted by the rate-limiting step model, depends on the concentrations of the GEF and GAP used: the synergy is strongest when both steps—GDP release and GTP hydrolysis—are rate-limiting, which occurs at higher concentrations of GEF and GAP (Fig. 6B). Conversely, the synergy diminishes when these steps are not rate-limiting, i.e., at lower effector concentrations (Fig. 6A). At which GEF and GAP concentrations do GDP release and GTP hydrolysis become rate limiting? When either step becomes rate-limiting, we expect the overall GTP hydrolysis rate $K$ to saturate (Fig. 6C). We conducted experiments with only one effector present (Cdc42 + Cdc24, Cdc42 + Rga2): we do not observe saturation of $K$ with increasing Cdc24 concentration (Fig. 3B), suggesting that GDP release does not become rate-limiting within our tested concentration range. Although $K$ does saturate with full-length Rga2 (Fig. 4B), it does not with the GAP domain (Fig. 4C), suggesting that we also do not reach concentration regimes where GTP hydrolysis becomes rate-limiting. Thus, we expect the overall synergy due to rate-limiting steps to be small. In addition, in the rate-limiting step model, the maximal possible accelerations by the GEF and GAP are interconnected, i.e., the maximal acceleration through the GEF limits the maximal acceleration that can be achieved by the GAP (and vice versa). We analyzed whether the accelerations that we observed by the GEF and GAP fit this model (Appendix Supplementary Text 4). In brief, we experimentally observe significantly larger accelerations by the GEF and GAP alone than the rate-limiting step model allows for (Appendix Table S5), also suggesting that it cannot fully explain the synergy term in our data.

More importantly, we carried out GTPase assays with Cdc24 and the GAP domain (Fig. 5B) and observed that here the interaction term $K_{3,Cdc24,GAP}$ is close to zero (Fig. 5D; Table 3). If the rate-limiting step model fully accounts for the synergy term in

our data, $K_{3,Cdc24,GAP}$ would be expected to be close to $K_{3,Cdc24,Rga2}$, given the comparable rates of the GAP domain and Rga2$_{(I)}$. This strongly supports that $K_{3,Cdc24,Rga2}$ cannot be explained by the rate-limiting step model alone and points to a protein-specific synergy between Cdc24 and Rga2. We suspect that $K_{3,Cdc24,GAP}$ is close to, but not exactly, zero, due to synergistic effects arising from rate-limiting steps in the GTPase cycle. However, because the concentrations of Cdc24 and GAP used are well outside the regimes where GDP release or GTP hydrolysis become rate-limiting, these effects contribute minimally to the overall synergy, resulting in a $K_{3,Cdc24,GAP}$ close to zero.

Taken together, we observe a positive interaction term $K_{3,Cdc24,Rga2}$ in our data, which cannot be fully explained by rate-limiting steps in the GTPase cycle alone and instead points to a protein-specific synergy between Cdc24 and Rga2. Next, we used SEC–MALS to investigate Cdc24–Rga2$_{(I)}$ binding as a potential origin of the synergy (Fig. 7): We did not observe Cdc24–Rga2 heterodimers, as the Rga2 oligomer peak lacked Cdc24 (Fig. 7B, western blot). However, the Cdc24 peak shifted towards a higher molecular weight (Fig. 7A), suggesting that an Rga2 fragment binds to Cdc24. This implies weak binding, as no Cdc24–Rga2 dimers or oligomers survived the flow under which SEC–MALS is performed. We analyzed the SEC–MALS fractions using several SDS–PAGE staining techniques and Western blotting. We detected a ~25 kDa fragment of Rga2 in the Cdc24 peak, visible only with SYPRO Ruby staining (Fig. 7B, SYPRO Ruby (blue arrow)), revealing it lacks both the N-terminal His-tag (detected via anti-His western blotting) and tryptophan residues (detected via stain-free SDS–PAGE). We analyzed the Rga2 sequence to determine if the lack of tryptophan could indicate which Rga2 domain binds to Cdc24, but all tryptophan-devoid Rga2 fragments are at least 25 kDa, meaning any of them could match the fragment observed on gel (Appendix Fig. S9). Therefore, the binding region remains unclear. We also conducted GTPase assays with Cdc24 and Rga2$_{(II)}$, which again showed synergy, although 40% reduced (Table 3; Appendix Fig. S11a–e). We suspect that the C-terminal Flag-tag of Rga2$_{(II)}$, which is not present in Rga2$_{(I)}$, may weaken Cdc24-Rga2 binding, resulting in a reduced synergy between the proteins. This would imply that Rga2's C-terminus is involved in Cdc24-Rga2 binding.

In conclusion, our data reveal a GEF–GAP synergy that is specific to Rga2 and Cdc24, absent in assays containing Cdc24 and the GAP domain. This synergy appears to arise from weak Cdc24-Rga2 binding, through which Cdc24 might alleviate the self-inhibition of oligomeric Rga2 (Fig. 8).

## Discussion

Synergy in GTPase regulation can arise through distinct mechanisms. One synergy arises from the acceleration of rate-limiting steps in the GTPase cycle by GEFs and GAPs: when both GDP release and GTP hydrolysis are rate-limiting, the combined action of GEFs and GAPs can enhance the cycling rate more than either effector

**Table 3. Cdc42–effector protein X interaction rates $k_{3,X_1}$, $k_{3,X_2}$, and $k_{3,X_1,X_2}$.**

| | Effector Protein $X_1$ | Effector Protein $X_2$ | $k_{3,X_1}$ [µM$^{-3}$ h$^{-1}$] | $k_{3,X_1}$ std. err. | $k_{3,X_2}$ [µM$^{-2}$ h$^{-1}$] | $k_{3,X_2}$ std. err. | $k_{3,X_1,X_2}$ [µM$^{-4}$ h$^{-1}$] | $k_{3,X_1,X_2}$ std. err. |
|---|---|---|---|---|---|---|---|---|
| Pooled est. ($n = 7$) | Cdc24 | Rga2$_{(I)}$ | 15.126 | 0.736 | 0.131 | 0.015 | 86.961 | 6.055 |
| Pooled est. ($n = 7$) | Cdc24 | Rga2$_{(II)}$ | 13.001 | 2.961 | 0.481 | 0.090 | 51.702 | 4.827 |
| Pooled est. ($n = 11$) | Cdc24 | GAP domain | 14.895 | 1.579 | 0.188 | 0.012 | 0.842 | 1.674 |

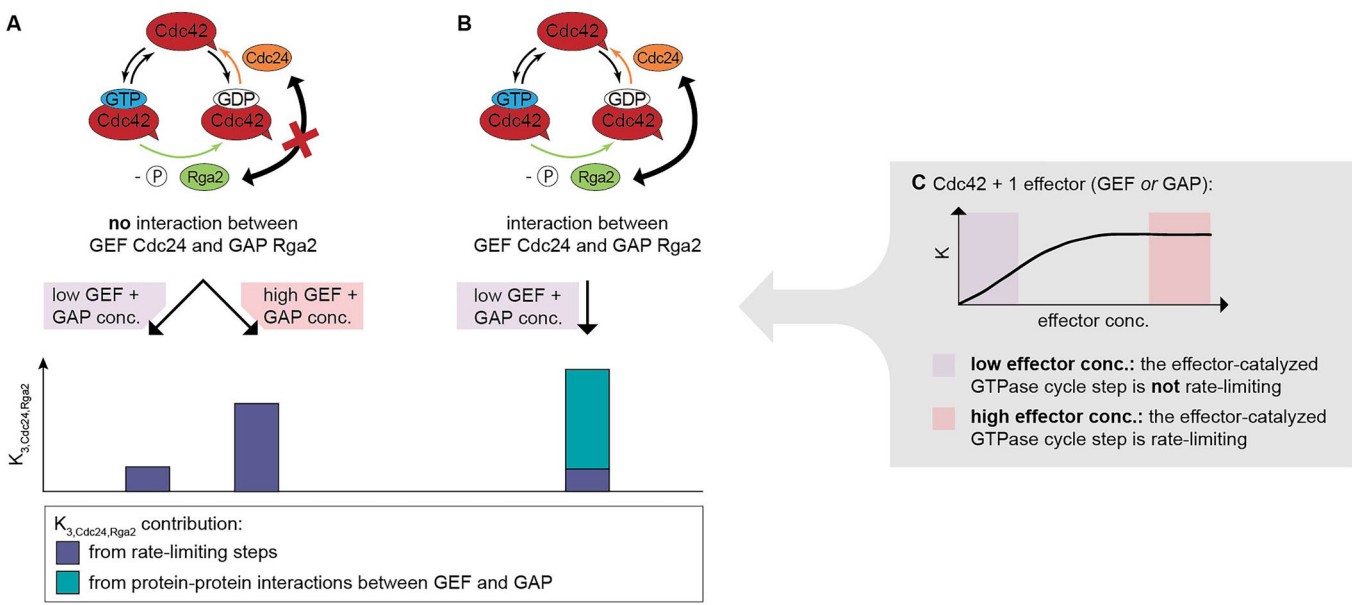

**Figure 6. GEF–GAP synergies can originate from rate-limiting steps and specific GEF–GAP interactions.**

(**A**) In the absence of an GEF–GAP interaction, synergy arises solely from rate-limiting steps. At low effector concentrations, the GEF- and GAP-catalyzed GTPase cycle steps are not in the rate-limiting regime (see (**C**)), resulting in a small synergy. At high effector concentrations, the GEF- and GAP-catalyzed GTPase cycle steps are in the rate-limiting regime, resulting in a larger synergy (see (**C**)). (**B**) In the presence of a specific GEF–GAP interaction, synergy can additionally arise from protein-protein interactions. At low effector concentrations, the contribution of the synergy from protein-protein interactions dominates, as the contribution from rate-limiting steps in this concentration regime is low. (**C**) GEF and GAP concentrations define rate-limiting step regimes: at low effector concentrations (purple regime), the effector-catalyzed step is not rate-limiting, and increases in effector concentrations lead to a higher overall GTPase hydrolysis rate $K$. At high effector concentrations (red regime), the effector the effector-catalyzed step is rate-limiting, thereby capping the overall GTPase hydrolysis rate $K$.

alone. Our data points to an additional form of synergy, which is based on protein-protein interactions: we observed a synergistic effect between Cdc24 and Rga2, which is absent in data where Cdc24 is combined with the isolated GAP domain. Our data suggest a novel functional coupling between GEFs and GAPs that arises due to protein-specific interactions between Cdc24 and Rga2. This underscores the importance of considering the molecular context when dissecting GTPase regulatory networks.

Our findings indicate that oligomerization is a critical factor shaping the GEF and GAP activities of Cdc24 and Rga2 (Fig. 8): Cdc24's GEF activity increases non-linearly with concentration, and data from an oligomerization-reduced mutant suggest that this non-linearity may result from the release of self-inhibition upon oligomerization (Shimada et al, 2004), similar to the activity-enhancing effect of Bem1 (Rapali et al, 2017). Rga2 also appears to self-inhibit, as indicated by the saturation of its activity observed in the data. Interestingly, Rga2's self-inhibition might originate from oligomerization and be alleviated by Cdc24. We suspect that this is the source of the GEF–GAP synergy, consistent with our finding

that Cdc24 and the GAP domain lack synergy, as the GAP domain's activity does not saturate (i.e., does not self-inhibit). Taken together, our data indicate that oligomerization has opposing effects on the GEF Cdc24 and the GAP Rga2; while Cdc24 overcomes self-inhibition, Rga2 self-inhibits. Subsequently, Rga2 relies on Cdc24 to overcome its self-inhibition, suggesting a functional connection between GAP activity and the GEF.

The emergence of Rga2's dependence on Cdc24 to release its self-inhibition could be the result of epistasis - a phenomenon where the effect of one gene masks or modifies the effect of another gene: Cdc24 is an essential gene product in *S. cerevisiae* and highly conserved in the fungal tree (Diepeveen et al, 2018), making it a persistent player in the polarity network. Mutations in Rga2 that result in self-inhibition would be permitted and masked by Rga2's synergistic effect with Cdc24, and could emerge without reducing the organism's fitness. Rga2 is only one of four GAPs of Cdc42 (among Rga1, Bem2, and Bem3). It was suggested that each GAP plays a distinct role in Cdc42 regulation, of which the level of GAP activity could be a part of (Smith et al, 2002). The role of GAP

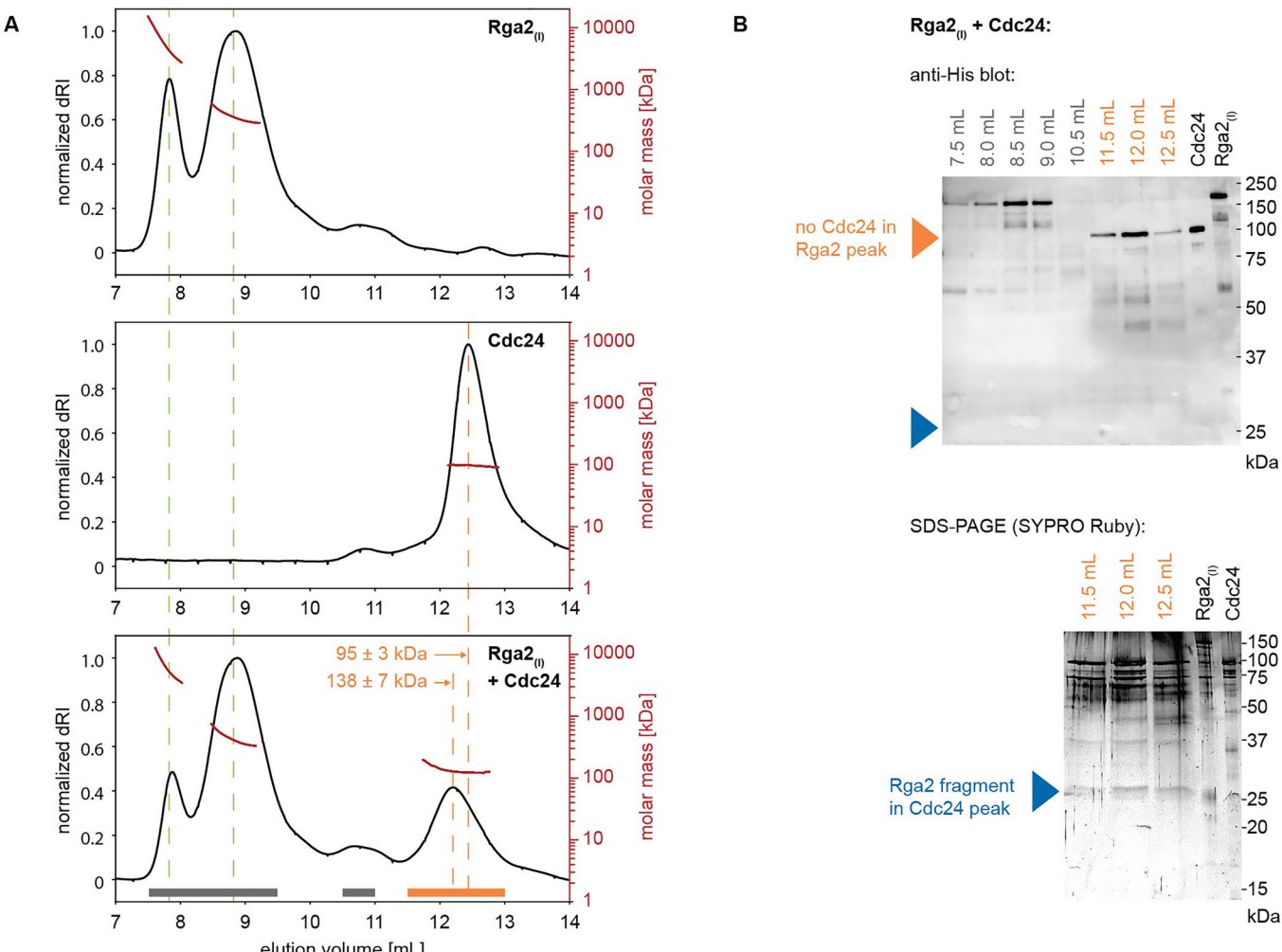

**Figure 7. The GEF Cdc24 and GAP Rga2 bind through weak interactions.**

Size-exclusion profile and MALS analysis (**A**) and SDS–PAGE/western blot analysis of SEC–MALS elution fractions (**B**) of Rga2$_{(l)}$, Cdc24, and Rga2$_{(l)}$ - Cdc24 mixtures: In SEC–MALS experiments with Cdc24, Rga2$_{(l)}$, and their mixtures, full-length Cdc24 is not present in Rga2 peaks (western blot: orange arrow). Instead, binding occurs with an Rga2$_{(l)}$ fragment (~25 kDa, SDS–PAGE: blue arrow), which shifts the Cdc24 peak towards larger molecular weights. This Rga2 fragment is devoid of tryptophan and lacks a His-tag (western blot: blue arrow). Source data are available online for this figure.

oligomerization, self-inhibition, and interaction with the GEF and other regulatory factors could be other distinguishing factors. In vitro analysis of other GAPs would help to clarify if and how these factors contribute to different roles of GAPs in *S. cerevisiae*.

Our data exemplifies non-linearities of Cdc42 GTPase cycle regulation: (1) the overall GTP hydrolysis rate of Cdc42 increases quadratically with Cdc24 concentration, and (2) the GEF Cdc24 and GAP Rga2 exhibit synergy. Both non-linearities could contribute to establishing polarity through creating regimes of high and low Cdc42 activity: (1) Temporal regulation of Cdc42 activity: In vivo, the timed release of Cdc24 from the nucleus (thus suddenly increasing the effective Cdc24 concentration) is known to be part of the polarity trigger (Shimada et al, 2000). We suspect that the nonlinear increase of the overall GTP hydrolysis rate of Cdc42 with Cdc24 concentration is a mechanistic element of Cdc24's function in vivo: once Cdc24 gets released from the nucleus, the GTPase cycling speed of Cdc42 increases strongly and suddenly (due to its nonlinear dependence on Cdc24

concentration). If Cdc24 additionally releases Rga2's self-inhibition and increases its GAP activity, Cdc24 further boosts Cdc42 GTPase cycling through activating Rga2. Timed by the release of Cdc24 from the nucleus, cells can quickly transition from a regime of low GTPase activity of Cdc42 (before the release of Cdc24) to a regime of high GTPase activity of Cdc42 (after the release of Cdc24). This sudden change in Cdc42's GTPase cycling speed could be part of the trigger that initiates polarity establishment. (2) Spatial regulation of Cdc42 activity: The synergistic regulation of Cdc42's GTPase activity through GEFs and GAPs is a resourceful and advantageous way of regulation; if regulatory factors have a synergistic interplay, wide ranges of up-regulation can be achieved through a small number of components. This synergy also implies that Cdc42 has a significantly higher GTPase activity at the polarity spot, where it is surrounded by many effector proteins that also regulate each other. We suspect the strong up-regulation at the site of bud emergence and the rather low baseline activity at other sites to have a cellular purpose, and imagine it is contributing to Cdc42 accumulation.

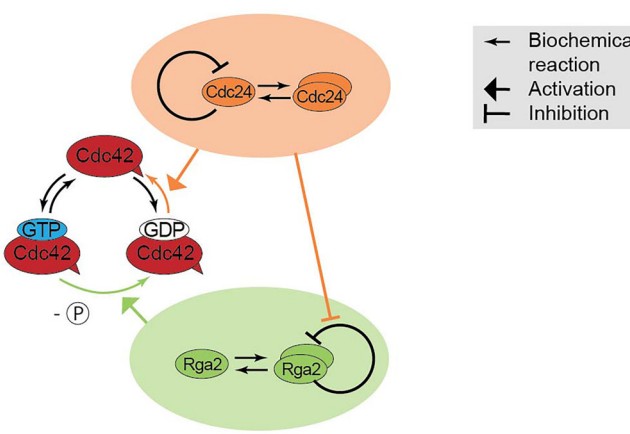

**Figure 8. Oligomerization-driven regulation of Cdc42 GTPase activity.**

Speculative model of Cdc42 GTPase activity regulation through the GEF Cdc24 and GAP Rga2: Both Cdc24 and Rga2 form oligomers. Cdc24's self-inhibition is released upon oligomerization, increasing its GEF activity. In contrast, Rga2 self-inhibits upon oligomerization. Together, Cdc24 and Rga2 synergize due to Cdc24 alleviating the self-inhibition of oligomeric Rga2.

Beyond its impact on our understanding of the yeast system, our findings may also apply to other eukaryotes: Cdc42 is highly conserved among eukaryotes and plays a central role in polarity establishment in many of these (Nelson, 2003; Etienne-Manneville, 2004; Thompson, 2013; Diepeveen et al, 2018). We imagine that the principles of its regulation are also conserved there, and propose that synergies between GEFs and GAPs also occur in these systems.

## Methods

### Reagents and tools table

| Reagent/resource | Reference or source | Identifier or catalog number |
|---|---|---|
| **Experimental models** | | |
| BL21 DE::3 pLysS cells | Thermo Fisher Scientific | C6060-10 |
| **Recombinant DNA** | | |
| pRV007 | Cdc42, this study | See Appendix Fig. S12 |
| pRV014 | Rga2$_{(I)}$, this study | See Appendix Fig. S12 |
| pRV110 | Rga2$_{(II)}$, this study | See Appendix Fig. S12 |
| pRV138 | GAP domain (aa 797–981 from Rga2 (Smith et al, 2002)), this study | See Appendix Fig. S12 |
| pDM272 | Cdc24, received from D. McCusker (University of Bordeaux) (Rapali et al, 2017) | See Appendix Fig. S12 |
| pRV136 | Cdc24-DH5 (incl. mutation F322A (Mionnet et al, 2008)), this study | See Appendix Fig. S12 |
| pRV137 | Cdc24-DH3 (incl. mutation L339A, E340A (Mionnet et al, 2008)), this study | See Appendix Fig. S12 |

| Reagent/resource | Reference or source | Identifier or catalog number |
|---|---|---|
| pET28a-His-mcm10-Sortase-Flag | Received from N. Dekker (TU Delft) | |
| **Antibodies** | | |
| His Tag Antibody, Mouse | Aviva Systems Biology | OAEA00010 |
| Goat Anti-Mouse | Aviva Systems Biology | OASA06620 |
| **Oligonucleotides and other sequence-based reagents** | | |
| PCR primers | This study | See Appendix Table S6 |
| **Chemicals, enzymes, and other reagents** | | |
| Casein | Sigma-Aldrich | cat. no. C7078 |
| BSA | Thermo Scientific | cat. no. 23209 |
| Ras (human) | EMD Millipore | cat. no. 553325 |
| cOmplete(TM) Protease Inhibitor Cocktail, EDTA free | Sigma-Aldrich | 000000011836145001 |
| Reagents, Enzyme Substrates, IPTG, Dioxane-Free | Thermo Fischer Scientific | 10849040 |
| Tween-20 | Sigma-Aldrich | P1379-100 |
| NP-40 alternative | Merck-Millipore | 492016-100 |
| Triton-X-100 | Sigma-Aldrich | X100-1L |
| 2-mercaptoethanol | Sigma-Aldrich | M6250 |
| 40% acrylamide solution | Bio-Rad | 1610148 |
| sodium dodecyl sulfate | Sigma-Aldrich | L3771 |
| 2,2,2-Trichloroethanol | Sigma-Aldrich | T54801 |
| ammonium persulfate | Sigma-Aldrich | A3678 |
| N,N,N',N'-tetramethyl ethylenediamine | Sigma-Aldrich | T22500 |
| isopropanol | Sigma-Aldrich | 34863 |
| Precision Plus Protein Unstained standard | Bio-Rad | 1610363 |
| Precision Plus Protein All Blue Standard | Bio-Rad | 1610373 |
| SimplyBlue SafeStain | Invitrogen | LC6065 |
| SYPRO Ruby protein gel stain | Invitrogen | S12000 |
| SilverQuest Staining Kit | Invitrogen | 45-1001 |
| blotting membrane (Trans-Blot Turbo Transfer Pack) | Bio-Rad | 1704156 |
| Immobilon signal enhancer | Millipore | WBSH0500 |
| SuperSignal West Pico Mouse IgG Detection Kit | Thermo Fischer Scientific | 34082 |
| **Software** | | |
| GTPase model | Available at Tschirpke et al (2023a) Described in Tschirpke et al (2024) | |

| Reagent/resource | Reference or source | Identifier or catalog number |
|---|---|---|
| **Other** | | |
| Lysis buffer: 50 mM Tris-HCl (pH=8.0), 1 M NaCl, 5 mM imidazole, 1 mM 2-mercaptoethanol, supplemented with EDTA-free Protease inhibitor cocktail (Roche) and 1 mM freshly prepared PMSF. | This study | |
| His-AC washing buffer: 50 mM Tris-HCl (pH=8.0), 1 M NaCl, 5 mM imidazole, 1 mM 2-mercaptoethanol | This study | |
| His-AC elution buffer: 50 mM Tris-HCl (pH=8.0), 100 mM NaCl, 500 mM imidazole, 1 mM 2-mercaptoethanol | This study | |
| SEC buffer: 50 mM Tris-HCl (pH=7.5), 100 mM NaCl, 10 mM MgCl₂, 1 mM 2-mercaptoethanol | This study | |
| GTPase Glo assay | Promega | V7681 |
| Compat-Able Protein Assay Preparation Reagent Kit | Thermo Fischer Scientific | 11861345 |
| BCA assay: Pierce BCA Protein Assay Kit | Thermo Fischer Scientific | 10750985 |
| high-pressure homogenizer (French press cell disruptor) | CF1 series Constant Systems | |
| HisTrap™ Excel column | Cytiva | 17524801 |
| HiPrep 16/60 Sephacryl S-300 HR column | Cytiva | 17116701 |
| Superdex 200 Increase 10/300 GL column | Cytiva | 28990944 |
| Amicon® Ultra 4 mL centrifugal filters 10KCO | Merck | UFC801024 |
| Durapore Membrane Filter 0.1 μM | Millipore | VVLP04700 |
| Innova 2300 platform shaker | New Brunswick Scientific | |
| 384-well plates | Corning | 3572 |
| Synergy HTX plate reader | BioTek | |
| high-performance liquid chromatography (HPLC) unit | 1260 Infinity II, Agilent | |
| UV detector | 1260 Infinity II VWD, Agilent | |
| 8-angle static light scattering detector | DAWN HELEOS 8 +; Wyatt Technology | |
| refractometer | Optilab T-rEX; Wyatt Technology | |

| Reagent/resource | Reference or source | Identifier or catalog number |
|---|---|---|
| Mini-PROTEAN® Spacer Plates with 1.0 mm Integrated Spacers | Bio-Rad | 1653311 |
| Mini-PROTEAN® Short Plates | Bio-Rad | 1653308 |
| Mini-PROTEAN® Comb, 12-well, 20 µl | Bio-Rad | 4560015 |
| PowerPac Basic Power Supply | Bio-Rad | |
| ChemiDoc MP | Bio-Rad | |
| Trans-Blot Turbo Transfer System | Bio-Rad | |
| ChemiDoc MP | Bio-Rad | |

## Plasmid construction

Genes of interest (Cdc42, Rga2) were obtained from the genome of *Saccharomyces cerevisiae* W303 and were amplified through PCR. The target vector (pET28a-His-mcm10-Sortase-Flag, received from N. Dekker (TU Delft) and based on pBP6 (Douglas and Diffley, 2016)) was also amplified through PCR. In addition, each PCR incorporated small homologous sequences needed for Gibson assembly (Gibson et al, 2009). After Gibson assembly, the resulting mixture was used to transform chemically competent Dh5α and BL21 DE::3 pLysS cells and plated out onto a Petri dish containing Lysogeny broth agar and the correct antibiotic marker.

An overview of the plasmids used is given in the "Reagents and Tools Table" accompanying this publication. Construction routes of plasmids and primers used for each PCR are shown in Appendix Fig. S12 and Appendix Table S6. The amino acid sequences of the proteins used in this publication are stated in Appendix Supplementary Text 5 and their design is discussed in detail in (preprint: Tschirpke et al, 2023b).

## Buffer composition

If not mentioned otherwise, buffers are of the composition stated in the "Reagents and Tools Table", accompanying this publication.

## Protein expression and purification

**Cdc42 (pRV007)** was expressed in Bl21::DE3 pLysS cells. Cells were grown in Lysogeny broth at 37 °C until an $OD_{600}$ of 0.7. The expression was induced through the addition of 1.0 mM IPTG, after which cells were grown for 3 h at 37° C. Cells were harvested through centrifugation. Cell pellets were resuspended in lysis buffer and lysed with a high-pressure homogenizer (French press cell disruptor, CF1 series Constant Systems) at 4 °C, using 5–10 rounds of exposing the sample to pressurization. The cell lysate was centrifuged at 37,000×*g* for 30 min, and the supernatant was loaded onto a HisTrap™ Excel column (Cytiva). After several rounds of washing with His-AC washing buffer, the protein was eluted in a gradient of His-AC washing buffer and His-AC elution buffer. The protein was dialyzed twice in SEC buffer. After the addition of 10%

glycerol, samples were flash-frozen in liquid nitrogen and kept at −80 °C for storage.

The expression and purification of **Cdc24 (pDM272)** is, with modifications, based on the protocol described previously (Rapali et al, 2017): Cdc24 was expressed in Bl21::DE3 pLysS cells. Cells were grown in Lysogeny broth at 37 °C until an $OD_{600}$ of 0.7. The expression was induced through the addition of 0.2 mM IPTG, after which cells were grown for 18 h at 18 °C. Cells were harvested through centrifugation. Cell pellets were resuspended in lysis buffer and lysed with a high-pressure homogenizer (French press cell disruptor, CF1 series Constant Systems) at 4 °C, using 5–10 rounds of exposing the sample to pressurization. The cell lysate was centrifuged at 37,000×g for 30 min, and the supernatant was loaded onto a HisTrap™ Excel column (Cytiva). After several rounds of washing with His-AC washing buffer, the protein was eluted in a gradient of His-AC washing buffer and His-AC elution buffer. The sample was further purified by size-exclusion chromatography using SEC buffer and a HiPrep 16/60 Sephacryl S-300 HR (Cytiva) column. Fractions containing full-size protein were concentrated using Amicon®Ultra 4 mL centrifugal filters (Merck). After the addition of 10% glycerol, samples were flash-frozen in liquid nitrogen and kept at −80 °C for storage.

Cdc24 mutants **Cdc24-DH3 (pRV137)** and **Cdc24-DH5 (pRV136)** were expressed and purified in the same fashion as Cdc24 (pDM272), with the only modification that the Lysis buffer, His-AC washing buffer, and His-AC elution buffer were all supplemented with 0.1% Tween-20, 0.1% NP-40, and 0.1% Triton-X-100.

**Rga2**$_{(I)}$ **(pRV014)**, and **Rga2**$_{(II)}$ **(pRV110)** were expressed in Bl21::DE3 pLysS cells. Cells were grown in Lysogeny broth at 37 °C until an $OD_{600}$ of 0.7. The expression was induced through the addition of 0.2 mM IPTG, after which cells were grown for 18–24 h at 10 °C. Cells were harvested through centrifugation. Cell pellets were resuspended in lysis buffer (supplemented with 0.1% Tween-20, 0.1% NP-40, and 0.1% Triton-X-100) and lysed with a high-pressure homogenizer (French press cell disruptor, CF1 series Constant Systems) at 4 °C, using 5–10 rounds of exposing the sample to pressurization. The cell lysate was centrifuged at 37,000×g for 30 min, and the supernatant was loaded onto a HisTrap™ Excel column (Cytiva). After several rounds of washing with His-AC washing buffer (supplemented with 0.1% Tween-20, 0.1% NP-40, and 0.1% Triton-X-100), the protein was eluted in a gradient of His-AC washing buffer and His-AC elution buffer (both supplemented with 0.1% Tween-20, 0.1% NP-40, and 0.1% Triton-X-100). The sample was further purified by size-exclusion chromatography using SEC buffer and a HiPrep 16/60 Sephacryl S-300 HR (Cytiva) column. Fractions containing full-size protein were concentrated using Amicon®Ultra 4 mL centrifugal filters (Merck). After the addition of 10% glycerol, samples were flash-frozen in liquid nitrogen and kept at −80 °C for storage.

**GAP domain (pRV138)** was expressed in Bl21::DE3 pLysS cells. Cells were grown in Lysogeny broth at 37 °C until an $OD_{600}$ of 0.7. The expression was induced through the addition of 1.0 mM IPTG, after which cells were grown for 3 h at 37 °C. Cells were harvested through centrifugation. Cell pellets were resuspended in lysis buffer and lysed with a high-pressure homogenizer (French press cell disruptor, CF1 series Constant Systems) at 4 °C, using 5–10 rounds of exposing the sample to pressurization. The cell lysate was centrifuged at 37,000×g for 30 min, and the supernatant was loaded onto a HisTrap™ Excel column (Cytiva). After several rounds of washing with His-AC washing buffer, the protein was eluted in a

gradient of His-AC washing buffer and His-AC elution buffer. The sample was further purified by size-exclusion chromatography using SEC buffer and a HiPrep 16/60 Sephacryl S-300 HR (Cytiva) column. Fractions containing full-size protein were concentrated using Amicon®Ultra 4 mL centrifugal filters (Merck). After the addition of 10% glycerol, samples were flash-frozen in liquid nitrogen and kept at −80 °C for storage.

**Casein, bovine serum albumin (BSA), and Ras** were purchased. Casein (Sigma-Aldrich, cat. no. C7078) was dissolved in SEC buffer. BSA (23209, Thermo Scientific) was dialyzed twice in SEC buffer. Ras (human) (EMD Millipore, cat. no. 553325) was diluted in SEC buffer.

All proteins are shown on SDS–PAGE in Appendix Fig. S13. To determine protein concentrations, samples were treated with Compat-Able Protein Assay Preparation Reagent Kit (Thermo Fisher Scientific) and analyzed using a BCA assay (Pierce BCA Protein Assay Kit, Thermo Fisher Scientific).

### Note on His-affinity chromatography

HisTrap™ Excel column (Cytiva) columns bought in 2020 or later required a higher amount of imidazole in the lysis and washing buffer, as stated in the "Reagents and Tools Table" accompanying this publication, as indicated by the recommendation "use 20–40 mM imidazole in sample and binding buffer for highest purity" on the column package. For these columns, the amount of imidazole in the lysis and His-AC washing buffer was increased to 50 mM.

An overview of the plasmids used is given in the "Reagents and Tools Table" accompanying this publication.

## GTPase assay

GTPase activity was measured using the GTPase-Glo™ assay (Promega), following the steps described in the assay manual and in (Tschirpke et al, 2024). In brief, 5 μL protein in SEC buffer (50 mM Tris-HCl (pH = 7.5), 100 mM NaCl, 10 mM $MgCl_2$, 1 mM 2-mercaptoethanol) was mixed with 5 μL of a 2× GTP-solution (10 μM GTP, 50 mM Tris-HCl (pH = 7.5), 100 mM NaCl, 10 mM $MgCl_2$, 1 mM 2-mercaptoethanol (Sigma-Aldrich), 1 mM dithiothreitol in 384-well plates (Corning) to initiate the reaction. The reaction mixture was incubated for 60–100 min at 30 °C on an Innova 2300 platform shaker (New Brunswick Scientific) (120 rpm), before the addition of 10 μL Glo buffer and another 30 min of incubation. The Glo buffer contains a nucleoside-diphosphate kinase that converts remaining GTP to ATP. The addition of 20 μL detection reagent, containing a luciferase/luciferin mixture, makes the ATP luminescent, which was read on a Synergy HTX plate reader (BioTek) in luminescence mode. Each data point shown corresponds to the average of 3–4 replica samples per assay.

This GTPase assay can be sensitive to small concentration differences, especially of effector proteins. To reduce the variability between assays and to increase comparability of different assay sets, 6× protein stocks were made using serial dilutions (with SEC buffer). The assays were conducted using the same 6× protein stocks within a few days, during which these stocks were kept at 4 °C. For each assay, equivalent volumes of 6× protein stocks were diluted to 2× mixtures (e.g., 10 μL Cdc42 + 20 μL SEC buffer, 10 μL Cdc42 + 10 μL effector protein 1 + 10 μL SEC buffer, 10 μL Cdc42 + 10 μL effector protein 1 + 10 μL effector protein 2, …). Incubation of 5 μL of this protein

mixture with 5 μL of 2× GTP solution, as described above, resulted in the concentrations stated in the figures.

We first assessed Cdc42 alone using several Cdc42 concentrations. We then examined the individual Cdc42–effector mixtures (two-protein assays), using one Cdc42 concentration and several effector concentrations. We then conducted assays using Cdc42 and two effectors. These three-protein assays contain wells with Cdc42, Cdc42 + effector 1, Cdc42 + effector 2, Cdc42 + effector 1 + effector 2, and additional "buffer" wells used for normalization. For feasibility, the three-protein assays include a reduced number of effector concentrations (in comparison to the two-protein assays).

## Fitting of GTPase assay data

The GTPase model, analysis procedure, and analysis code are described in detail in (Tschirpke et al, 2024) and available at (data ref.: Tschirpke et al, 2023a). Key aspects are summarized here:

The amount of remaining GTP correlates with the measured luminescence. Wells without protein ("buffer") were used for the normalization and represent 0% GTP hydrolysis:

$$[\text{GTP}]_{t_{term.}} = \left(\frac{\text{Lum.protein}}{\text{Lum.buffer}}\right)$$

Wells where no GTP was added showed luminescence values corresponding to <1% remaining GTP. Given the small deviation to 0%, and that GTPase reactions of protein mixtures leading to <5% remaining GTP were excluded from further analysis (Tschirpke et al, 2024), we did not normalize the data using luminescence values corresponding to 0% GTP.

Reactions were carried out with three to four replicates (wells) per assay, and the average ("Lum.") and standard error of the mean ("ΔLum.") of each set was used to calculate the amount of remaining GTP at the time of reaction termination and the error of each set:

$$\Delta \text{ remaining GTP} = \sqrt{\left(\frac{\Delta\text{Lum.protein}}{\text{Lum.protein}}\right)^2 + \left(\frac{\Delta\text{Lum.buffer}}{\text{Lum.buffer}}\right)^2}\frac{\text{Lum.protein}}{\text{Lum.buffer}}$$

The data was fitted using a GTPase activity model (described in brief in Appendix Supplementary Text 1 and in detail in (Tschirpke et al, 2024)) where the GTP decline occurring during the GTPase reaction is approximated with an exponential:

$$[\text{GTP}]_t = [\text{GTP}]_{0h}\exp(-Kt)$$
$$\text{using}[GTP]_{0h} = 1$$
$$\text{and } K = k_1 c_{corr}[\text{Cdc42}] + k_2(c_{corr}[\text{Cdc42}])^2 + k_{3,X1}c_{corr}[\text{Cdc42}][\text{X1}]^n$$
$$+ k_{3,X2}c_{corr}[\text{Cdc42}][\text{X2}]^m + k_{3,X1,X2}c_{corr}[\text{Cdc42}][\text{X1}]^n[\text{X2}]^m$$

Here $K$ represents the overall GTP hydrolysis rate, $X1$ and $X2$ are effector proteins with $n$ and $m = \{1, 2\}$, and $c_{corr}$ is a variable used to map all factors that lead to variations between GTPase assays onto the Cdc42 concentration (Appendix Fig. S2. The pooled estimates of rates $k_1$, $k_2$, and $k_3$ were determined through weighting their standard error.

## Size-exclusion chromatography–multi-angle light scattering (SEC–MALS)

The oligomerization states of Cdc24, Rga2, the GAP domain, and mixtures thereof were estimated using analytical size-exclusion chromatography coupled to multi-angle light scattering (SEC–MALS). Purified protein samples were resolved on a

Superdex 200 Increase 10/300 GL column (Cytiva) connected to a high-performance liquid chromatography (HPLC) unit (1260 Infinity II, Agilent) running in series with an online UV detector (1260 Infinity II VWD, Agilent), an 8-angle static light scattering detector (DAWN HELEOS 8 +; Wyatt Technology), and a refractometer (Optilab T-rEX; Wyatt Technology). Experiments were performed using SEC buffer supplemented with 0.02% sodium azide and filtered through 0.1 μM pore filters (Durapore Membrane Filter 0.1 μM, Millipore). Protein samples were prepared at a total volume of 60 μL and final concentrations of 5.7–7.1 μM. After mixing, samples were incubated on ice for 30 min and then spun down at 21,000×g for 10 min (Eppendorf Centrifuge 5424 R). In total, 50 μL of protein sample was injected onto the column and eluted using SEC buffer supplemented with 0.02% sodium azide at 0.5 mL/min. Peak fractions were collected. 1 mg/mL monomeric BSA (Albumin monomer bovine, Sigma-Aldrich) was used as a reference for mass calibration. All experiments were performed in triplicate.

## SDS–PAGE

SDS–PAGE gels (12–15% acrylamide) were prepared freshly. In brief, a solution of 375 mM Tris-HCl (pH = 8.8), 30–37.5 v/v% 40% acrylamide solution (Bio-Rad), 0.2 w/v% sodium dodecyl sulfate (Sigma-Aldrich), 0.5 v/v% 2,2,2-Trichloroethanol (Sigma-Aldrich), 0.1 w/v% ammonium persulfate (Sigma-Aldrich), and 0.1 v/v% N,N,N',N'-tetramethyl ethylenediamine (Sigma-Aldrich) was prepared and casted into 1.00 mm mini-protean glass plates (Bio-Rad), filling them up to 80%. To protect the gel surface from drying, a layer of isopropanol (Sigma-Aldrich) was added. The gel was allowed to solidify for 20 min, after which the isopropanol layer was removed. A solution of 155 mM Tris-HCl (pH = 6.5), 10 v/v% 40% acrylamide solution (Bio-Rad), 0.2 w/v% sodium dodecyl sulfate (Sigma-Aldrich), 0.1 w/v% ammonium persulfate (Sigma-Aldrich), and 0.1 v/v% N,N,N',N'-tetramethyl ethylene-diamine (Sigma-Aldrich) was prepared and added to the existing gel layer, after which a well comb (Bio-Rad) was added. The gel was allowed to solidify for 20 min.

Protein samples were mixed with SDS loading buffer (Laemmli buffer, (Laemmli, 1970)) and kept for 5 min at 95 °C before loading onto the gels. Precision Plus Protein Unstained standard (Bio-Rad) (and Precision Plus Protein All Blue Standard (Bio-Rad) in case of Western blot analysis) was used as a protein standard. Gels were run for 5 min at 130 V followed by 55 min at 180 V (PowerPac Basic Power Supply (Bio-Rad)).

If not stated otherwise, imaging was done on a ChemiDoc MP (Bio-Rad) using the "Stain-free gels" feature and automatic exposure time determination. In some instances, gels were stained using SimplyBlue SafeStain (Invitrogen), SYPRO Ruby protein gel stain (Invitrogen), or SilverQuest Staining Kit (Invitrogen) following the basic steps of the accompanying manual. Imaging was done on a ChemiDoc MP (Bio-Rad) using automatic exposure time determination.

## Western blotting

After SDS–PAGE, the sample was transferred from the SDS–PAGE to a blotting membrane (Trans-Blot Turbo Transfer Pack, Bio-Rad) using the "Mixed MW" program of the Trans-Blot Turbo Transfer System (Bio-Rad). The blotting membrane was incubated with

Immobilon signal enhancer (Millipore) at room temperature for 1 h. The blotting membrane was incubated with primary antibody (His Tag Antibody, Mouse (OAEA00010, Aviva Systems Biology) (dilution: 1:4000)), diluted in Immobilon signal enhancer, at room temperature for 1 h. It was washed thrice with TBS-T (10 mM Tris-HCl (pH = 7.5), 150 mM NaCl, 0.1 v/v% Tween-20 (Sigma-Aldrich)). For each washing step, the blotting membrane was incubated with TBS-T at room temperature for 20 min. The blotting membrane was incubated with secondary antibody (IgG2b Antibody HRP-conjugated (Goat Anti-Mouse) (OASA06620, Aviva Systems Biology) (dilution: 1:1000)), diluted in Immobilon signal enhancer, at room temperature for 1 h, after which it was again washed thrice with TBS-T. SuperSignal West Pico Mouse IgG Detection Kit (Thermo Scientific) was used for activation. Imaging was done on a ChemiDoc MP (Bio-Rad) using the "Chemi Sensitive" feature and automatic exposure time determination.

## Data availability

The data underlying this publication are openly available at https://data.4tu.nl at https://doi.org/10.4121/56535d83-8a7a-4367-9ae5-f0d56b51161f (CC BY-SA 4.0) (data ref.: Tschirpke et al, 2025).

The source data of this paper are collected in the following database record: biostudies:S-SCDT-10_1038-S44319-026-00695-7.

## Peer review information

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

## Acknowledgements

We thank R van der Valk and C. de Agrela Pinto for experimental assistance, F Shamsi and S Farooq for their pioneering work on GTPase activity by Cdc42 and its regulators, and M Depken for discussions and careful reading of the manuscript. We thank D McCusker (University of Bordeaux) for the plasmid pDM272 and N. Dekker (TU Delft) for the plasmid pET28a-His-mcm10-Sortase-Flag. L Laan gratefully acknowledges funding from the European Research Council under the European Union's Horizon 2020 research and innovation programme (grant agreement 758132) and funding from the Netherlands Organization for Scientific Research (Nederlandse Organisatie voor Wetenschappelijk Onderzoek) through a Vidi grant (016.Vidi.171.060). S Tschirpke gratefully acknowledges funding from the Kavli Synergy Post-doctoral Fellowship program of the Kavli Institute of Nanoscience Delft.

## Author contributions

**Sophie Tschirpke**: Conceptualization; Resources; Data curation; Formal analysis; Validation; Investigation; Visualization; Methodology; Writing—original draft; Writing—review and editing. **Werner K-G Daalman**: Conceptualization; Software; Formal analysis; Validation; Investigation; Methodology. **Frank van Opstal**: Formal analysis; Investigation. **Liedewij Laan**: Conceptualization; Supervision; Funding acquisition; Validation; Investigation; Project administration; Writing—review and editing.

Source data underlying figure panels in this paper may have individual authorship assigned. Where available, figure panel/source data authorship is listed in the following database record: biostudies:S-SCDT-10_1038-S44319-026-00695-7.

## Disclosure and competing interests statement

The authors declare no competing interests.

