## [Peer Review File · EMBO Reports]

Oligomerization-dependent and synergistic regulation of Cdc42 GTPase cycling by a GEF and a GAP

Sophie Tschirpke, Werner Karl-Gustav Daalman, Frank van Opstal, and Liedewij Laan

Corresponding author(s): Liedewij Laan (l.laan@tudelft.nl)

Review Timeline:

Transfer Date:	19th Sep 23
Editorial Decision:	9th Oct 23
Appeal Received:	8th Oct 24
Editorial Decision:	23rd Dec 24
Revision Received:	13th Jun 25
Editorial Decision:	22nd Jul 25
Revision Received:	27th Oct 25
Accepted:	18th Dec 25

Transaction Report: This manuscript was transferred to EMBO reports following peer review at Review Commons.

**Review
COMMONS**

Review #1

1. Evidence, reproducibility and clarity:

Evidence, reproducibility and clarity (Required)

This study would be much convincing if additional line of eukaryotic cells can be used to demonstrate the GEF-GAP synergy tis important for cell physiology. In addition, it would be best to demonstrate the spatiotemporal interaction of GEF-GAP using high-resolution live cell imaging.

2. Significance:

Significance (Required)

The revised study would provide first line evidence that GEF-GAP synergy to be general regulatory property in eukaryotic kingdom.

3. How much time do you estimate the authors will need to complete the suggested revisions:

Estimated time to Complete Revisions (Required)

(Decision Recommendation)

Between 1 and 3 months

No

Review #2

1. Evidence, reproducibility and clarity:

Evidence, reproducibility and clarity (Required)

The study entitled, "The GEF Cdc24 and GAP Rga2 synergistically regulate Cdc42 GTPase cycling" by Tschirpke et al., uses an in vitro GTPase assay to examine the GTPase cycle of Cdc42 in combination with its GEF and GAP effectors. The authors find that the Cdc24 GEF activity scales non-linearly with its concentration and the GAP Rga2 has substantially weaker effect on stimulating Cdc42 GTPase activity. Not surprisingly, the combined addition of Cdc24 and Rga2 lead to a substantial increase in Cdc42 GTPase activity.

****Referees cross-commenting****

In Zheng, Y., Cerione, R., and Bender, A. (1994) J. Biol. Chem. 269: 2369-2372 (Fig. 3C), the authors show that Cdc24 combined with the GAP Bem3 lead to a large synergy in boosting Cdc42 GTPase activity.

2. Significance:

Significance (Required)

There is very little new information in this manuscript. Previous studies (Rapali et al. 2017) have shown that the scaffold protein Bem1 enhances the GEF activity of Cdc24. It is expected that the reconstitution of a GEF and GAP protein promote the GTPase cycle and indeed Zheng et al. (1994) showed that that Cdc24 combined with the GAP Bem3 lead to a large synergy in boosting Cdc42 GTPase activity. Hence the only potentially interesting finding in this work is that, in solution Cdc24 activity scales non-linearly with its concentration. However as this GEF and Cdc42 are associated with the membrane, the relevance of solution studies are less clear and furthermore the mechanistic basis for the non-linearity is not explored in detail. Given the limited new information from this work, the findings are, in their current form, too preliminary.

3. How much time do you estimate the authors will need to complete the suggested revisions:

Estimated time to Complete Revisions (Required)

(Decision Recommendation)

More than 6 months

4. Review Commons values the work of reviewers and encourages them to get credit for their work. Select 'Yes' below to register your reviewing activity at Web of Science Reviewer Recognition Service (formerly Publons); note that the content of your review will not be visible on Web of Science.

No

Review #3

1. Evidence, reproducibility and clarity:

Evidence, reproducibility and clarity (Required)

This work reports a biochemical analysis of the effects of a recombinant yeast GEF (Cdc24) and GAP (Rga2) on Cdc42 GTPase cycling in vitro. The central conclusion is that the GEF and GAP act "synergistically", which occurs "due to proteins enhancing each other's effects". By this they appear to mean that the GEF enhances the GAP's activity and vice versa. I was not persuaded that this is correct, and was confused by many aspects of the approach and interpretation, as outlined below.

1. GEF and GAP are expected to accelerate GTPase cycle synergistically even with no effect on each other's activity:

The Cdc42 GTPase cycle is understood to occur via distinct steps (GDP release, GTP binding, and GTP hydrolysis): GDP release and GTP hydrolysis are intrinsically slow steps that are accelerated by GEFs (GDP release) and GAPs (GTP hydrolysis). This fundamental biochemistry was established in the 1990s using biochemical assays that measure each step independently. Here instead the authors use an assay that measures [GTP] decline in a mix with 5 uM starting GTP, 1 uM Cdc42, plus or minus some amount of GEF or GAP. They assume exponential decline of [GTP] with time, yielding a cycling "rate". If that is so, then one would expect that added GEF would accelerate only the first step, leaving a slow GTP hydrolysis step that limits the overall cycling rate, while added GAP would accelerate only the last step, leaving a slow GDP release step that limits the overall cycling rate. Adding both together would speed up both steps, and should therefore "synergistically" accelerate cycling. This would be expected based on previous work and does not imply that GEF or GAP are affecting each other's action (except trivially by providing substrate for the next reaction). If the authors wish to demonstrate that something more complex is indeed happening, they need to use assays that directly measure the sub-reaction of interest, as done by prior investigators.

2. Model-based interpretation of the GTPase assay is poorly supported:

The assay employed measures overall GTP concentration with time. It is assumed (but not well documented-see below) that [GTP] declines exponentially, and that the rate constant for a particular condition can be fit by the sum of a series of terms that are linear or quadratic in the concentrations of Cdc42, GEF, and GAP. There is no theoretical derivation of this model from the elementary reactions, and the assumptions involved are not well articulated.

As discussed in point 1 above, one would expect that a GEF or GAP alone could only accelerate the cycle to a certain point, where the other (slow) reaction becomes rate limiting. But that does not appear to be true for their phenomenological model, where slow steps (small terms in the sum) will always be overwhelmed by fast steps. This is not the traditional understanding of how GTPases operate.

3. Data that do not conform to expectation are not explained:

Strangely, the data (as interpreted by the model assumptions) also appear inconsistent with the expectation of rate-limiting steps. GEF addition (alone) is said to accelerate cycling 100-fold, while GAP addition (alone) accelerates it 2-fold. But that would seem to imply that GDP release takes up >99% of the basal cycle (so accelerating that step alone reduces cycling time 100-fold), while GTP hydrolysis takes up >50% of the basal cycle (so accelerating that step alone reduces cycling time 2-fold). In the conventional understanding of GTPase cycles, these cannot both be true (as the steps would then add to >100% of the basal cycle). There is no attempt to reconcile these findings with previous work.

4. Lack of detailed timecourse data:

The decline in [GTP] with time is stated to be exponential, allowing extraction of an overall cycling "rate". But this claim is supported only weakly (S3 Fig. 1 uses only 3 timepoints, is not plotted on semi-log axis, and does not report fit to exponential vs other models) and only for the Cdc42-alone scenario: no data at all are presented to support exponential decline in reactions with GEF or GAP. Most assays seem to measure only a single timepoint, so extraction of a "rate" is very heavily influenced by the unsupported assumption of exponential decline. And if the decline is not exponential, it becomes extremely difficult to interpret what a single timepoint means.

5. Other issues with interpretation of the data:

(i) It is unclear why the authors chose to employ an assay that is much harder to interpret than the biochemical assays used by others. In biochemical studies, assays that report an output of multiple reactions are always harder to interpret than assays targeting a single

reaction. As well-established assays are available for each individual step in GTPase cycles, any conclusions must be supported using such assays.

(ii) The reported basal (and GEF/GAP-accelerated) rates are very slow, perhaps due to poor folding of recombinant proteins. This raises the possibility that much of the Cdc42 is inactive. If so, then accelerated GTP hydrolysis could come from increasing the active fraction of Cdc42, rather than catalyzing a specific step.

(iii) The GEF and GAP preparations include multiple partial degradation products and it is unclear whether the measured activities come from full-length proteins or more active fragments.

(iv) Cdc42 cycling is also accelerated by BSA and casein, suggesting that there are poorly understood aspects of the assay and that GEF and GAP actions may (like BSA and casein) involve non-canonical effects on Cdc42. As GEF and GAP are expected to interact better with Cdc42 than BSA or casein, these effects could dominate the observed changes in GTP levels.

(v) Cdc42-alone cycling assays are said to be reproducible. However, assays with added GEF/GAP/BSA/Casein yield rates that vary almost an order of magnitude between replicates. This poor reproducibility further reduces confidence in the findings.

(vi) It is unclear what timepoint was used for the different assays. 1.5 h at 30 degrees seems to be the standard here for the Cdc42-alone assays, but I assume that cannot be what was measured to assess GTP decline for GEF-containing assays as there would be very little GTP left at 1.5 h.

(vii) The graph reporting GEF activity is plotted only for $[GEF] < 0.2 \mu M$, but the rates used in the subsequent experiments are reported for mixtures with $1 \mu M$ GEF. The full range of GEF data should be plotted.

(viii) S8 Data with casein seems very noisy and it is no longer at all clear that the quadratic fit for $[Cdc24]$ is justified. Also, the symbol colors are very similar so it is hard to tell what data corresponds to what condition. The synergy between Cdc24 and Rga2 is also very noisy and the fits seem arbitrary.

(ix) It is disturbing that different Cdc42 constructs behave quite differently (S4). This suggests that protein behavior is influenced by the various added epitope tags and

protease cleavage sites (they also leave the C-terminal CAAX box rather than removing the AAX as would happen in vivo). These features raise the concern that these findings may not be directly relevant to the situation with endogenous yeast Cdc42. Of course, it is also the case that relevant Cdc42 biochemistry occurs with prenylated Cdc42 on membranes.

2. Significance:

Significance (Required)

The basic biochemistry of Cdc42 cycles was figured out about 30 years ago. However, those studies did not examine how combinations of Cdc42 regulators (as opposed to individual regulators) might interact to produce effects not expected from combining their individual actions. Recently, this combination approach did lead to interesting findings by Rapali et al. This approach is worthwhile and addresses a major question of interest to the broader field of GTPase biochemistry.

One main limitation of this study is technical: the main assay is less informative (though perhaps easier) than traditional assays, and it is unclear whether the recombinant proteins employed retain their normal activities. Another limitation is the model-based interpretation of the assay that does not include the potential for rate-limiting steps.

3. How much time do you estimate the authors will need to complete the suggested revisions:

Estimated time to Complete Revisions (Required)

(Decision Recommendation)

Cannot tell / Not applicable

4. Review Commons values the work of reviewers and encourages them to get credit for their work. Select 'Yes' below to register your reviewing activity at Web of Science Reviewer Recognition Service (formerly Publons); note that the content of your review will not be visible on Web of Science.

No

Review #4

1. Evidence, reproducibility and clarity:

Evidence, reproducibility and clarity (Required)

Summary

The GTPase cdc42 is a key determinant of yeast polarization. Its activity is amplified at the site of polarization through a poorly defined positive feedback mechanism, and depends on numerous GAPs regulating GTP hydrolysis and the GEF cdc24 that regulates GDP release. These components have previously been evaluated for their quantitative effects on the individual steps in the GTPase cycle that they modulate, but potential interactions between the cdc24 GEF and any GAP could not be examined based on these assays. The authors validate and employ a bulk assay of the total GTPase cycle based on GTP consumption to study the activities of and potential interactions between cdc24 and the GAP Rga2. Fitting their data to a mathematical model, they come to three central conclusions: (1) the activating activity of cdc24 to activate cdc42 GTPase activity is nonlinear, showing a quadratic relationship, (2) Rga2 shows a much lower activating activity that is linear at low levels before saturating, and (3) there is a strongly synergistic interaction between the activating activities of cdc24 and Rga2. Some hypotheses for the mechanistic bases of these findings are hypothesized, but not further investigated. Their conclusions are well supported by the data which appears to be of sufficient rigor.

Major comments

The three main conclusions of the manuscript are well supported by the data and associated modeling.

One unresolved issue is the discrepancy between the authors' conclusion that the non-linear activation by cdc24 is likely a result of oligomerization, whereas Mionnet et al 2008 reach the opposite conclusion. It seems that the authors wish to discount the Mionnet results because they used truncated constructs to test deficient oligomerization and an engineered construct to test induced oligomerization. If the authors are correct, then a relatively easy test would be to introduce the oligomerization deficient mutants defined by Mionnet into their full length construct and compare to wild type protein. While the authors' measured results don't depend on the offered mechanism and this experiment is therefore optional, their explanation is quite unsatisfying, especially since an experiment to resolve the difference is entirely feasible and not very strenuous.

Minor comments

The results in Fig S4 serve as assay validation, and this should be pointed out early in the Results section. I was initially concerned when the assay was described as based on consumption of GTP that a significantly diminished pool would alter the rate and thereby distort results, and being made aware of the S4 result would have alleviated that concern as I read further.

On page 4 and Fig S4 the authors mention several cdc42 constructs, some of which show linear activity curves and others slightly non-linear curves. I was unable to find where these constructs or their differences are discussed. The authors should also tell us if the construct used for the remaining experiments was one of the two shown in S4, or a different one.

It seems that in Fig 4 and Fig S8, some points are missing from the graphs. Were all concentrations for each condition not always assayed, or is some data omitted for some reason? For example, for the 0.125 microM Rga2 condition, only two points are shown vs 4 for some other conditions, and the two missing ones are expected to not be excluded by the >5% GTP remaining criterion.

In these graphs, a diamond symbol of slightly varying color is used for the different conditions. The different colors are hard to distinguish. Please use different shape symbols for the different conditions, and choose colors that are more distinct.

There are a few sentences that are of unclear meaning, for example on page 10, "It was suggested that each GAP plays a distinct role in Cdc42 regulation, of which the level of GAP activity could be a part of [Smith et al., 2002]." There are also typos and grammatical errors that should be fixed.

2. Significance:

Significance (Required)

The most novel and important finding is the strong synergy observed between cdc24 and Rga2 in activating cdc42 GTPase activity. This is undoubtedly an important mechanism underlying positive feedback in polarization. The measured non-linear activity of cdc24 alone is also quite important given that availability of cdc24 is thought to be a critical in vivo stimulus for polarization. However, the unexplained discrepancy between this result and that of Mionnet leaves one to wonder which result is more reliable. Only Mionnet attempts to directly test whether oligomerization is important in cdc24 activity.

The conclusions are of importance to a broad audience of cell biologists, though the lack of any mechanism for the synergy or the non-linearity of cdc24 activity somewhat diminishes significance.

Note that my expertise and that of my co-reviewer is in the biology, and while we are able to follow the contributions of the modeling, we do not have the expertise to critically evaluate for potential errors or weaknesses in the modeling itself.

3. How much time do you estimate the authors will need to complete the suggested revisions:

Estimated time to Complete Revisions (Required)

(Decision Recommendation)

Between 1 and 3 months

No

Dear Dr. Laan,

Thank you for the submission of your manuscript to EMBO Reports. I apologize for my delayed response, but I have now carefully assessed it and discussed it with the other editors. We noted that the referee's opinions and concerns were rather mixed, regarding the conceptual advance provided but also the proposed model on a synergistic mode of action of GEF and GAP. I have therefore further consulted with the referees and discussed the point-by-point response you provided. I am sorry to say that our conclusion from these considerations was that we cannot offer publication in EMBO Reports.

One of the concerns from referee 3 was that your data do not fit the rate limiting model. The textbook view of Ras-superfamily GTPase cycles is that of sequential steps in the cycle (GDP release, GTP binding, GTP hydrolysis), with GEFs accelerating only the GDP release and GAPs only the GTP hydrolysis. I would like to share with you the detailed comments from the referee as these might be helpful in further developing and strengthening your work (below my signature). The referee also pointed out that your model might overturn the textbook view and while we are in principle interested in studies that refute earlier data or broadly held beliefs, we also agree with the referee that in this case the data need to be strong and convincing. Based on these concerns we overall feel that a revision might not be productive and we have therefore decided not to proceed with the re-review of your manuscript. As noted above, EMBO Reports welcomes studies that refute broader claims or widely held beliefs, so in case you obtain data in the near future that would considerably strengthen your conclusions we would have no objections to reconsider a resubmission of your manuscript. I would like to stress though that such a manuscript would be treated as a new submission and would be evaluated again, also with respect to the literature and the novelty of your findings at the time of resubmission.

At this stage, however, I am sorry that I cannot bring better news and wish you success with the rapid publication of your manuscript.

Yours sincerely,

Referee feedback:

[...] GEFs accelerate only GDP release and GAPs accelerate only GTP hydrolysis. This implies that the remaining unaccelerated step(s) must limit the rate of overall cycling. My review pointed out that the data in the paper appear to be incompatible with that view. The authors agree, and say that they "do not opt for a rate-limiting model". But they never articulate what they do "opt for" in terms of the effects of GEF and GAP on each step of the cycle. I assume that the authors believe that each regulator (GEF and GAP) individually accelerates both GDP release and GTP hydrolysis.

If this is the model the authors propose, then:

- (i) They need to say so explicitly.
- (ii) They need to test their model using direct GDP release and GTP hydrolysis assays that are standard in the field.
- (iii) They should appreciate that this heterodox idea would overturn the textbook view, and therefore deserves thorough and convincing demonstration before moving on to examine the effects of combined GEF and GAP addition.

If they do not believe this, then:

- (i) They need to explicitly say what they think is going on.
- (ii) Whatever the model is needs to be tested with appropriate assays and controls.

Rev_Com_number: RC-2023-02060

New_manu_number: EMBOR-2023-58186V1

Corr_author: Laan

Title: The GEF Cdc24 and GAP Rga2 synergistically regulate Cdc42 GTPase cycling

General comments from the authors:

In the reviewed manuscript, we qualitatively and quantitatively investigated how the GEF Cdc24 and GAP Rga2 (alone and in combination) regulate GTPase cycling of *S. cerevisiae* Cdc42 in vitro. We found that Cdc24 and Rga2 have distinct concentration-dependent activities: Cdc24 increases Cdc42 GTPase cycling in a non-linear fashion and the effect of Rga2 saturates. We speculated that this is linked to Cdc24 and Rga2 oligomerization. Our data showed that Cdc24 and Rga2 together exhibit a synergy.

We appreciated the reviewers' comments, which prompted us to extend the scope of the manuscript in several key aspects:

1. We added GTPase assay data on an oligomerization-deficient Cdc24 mutant, strengthening the link between the observed non-linearity of Cdc24's GEF activity and Cdc24 oligomerization.
2. We added further insights onto the GAP activity of Rga2:
 - a. We show that Rga2 forms oligomers (Fig. 4a), which we suspect cause the saturation of Rga2's GAP activity due to self-inhibition upon oligomerization.
 - b. We added GTPase assay data using Rga2's GAP domain (Fig.4c), which – in contrast to Rga2 – does not show a saturation, supporting our hypothesis that the saturation effect of Rga2 is due to self-inhibition.
3. We confirm that the Rga2-Cdc24 synergy is protein specific through conducting GTPase assays with Cdc24 and Rga2's GAP domain (which do not exhibit synergy, Fig. 5b). This finding alleviates the reviewer's main concern that the synergy arises due to properties of a rate-limiting step model.
4. We added mechanistic insights on the origin of Cdc24-Rga2 synergy, showing that Rga2 and Cdc24 exhibit weak binding (Fig. 6, 7).

The reviewers' feedback greatly improved this manuscript, which now offers more in-depth details on the concentration- and oligomerization-dependent regulation of Cdc42 by Cdc24 and Rga2, along with stronger and more comprehensive evidence supporting Cdc24-Rga2 synergy and its underlying mechanism.

Reviewer #1 (Evidence, reproducibility and clarity (Required)):

This study would be much convincing if additional line of eukaryotic cells can be used to demonstrate the GEF-GAP synergy tis important for cell physiology. In addition, it would be best to demonstrate the spatiotemporal interaction of GEF-GAP using high-resolution live cell imaging.

Response from the authors:

The reviewer requests additional in vivo data to support our in vitro findings:

(1) The reviewer requests in vivo data showing that GEF-GAP synergy is important for cell physiology. We believe that in order to show GEF-GAP synergy in vivo, Cdc42 cycling rates would need to be measured in vivo. For that single-molecule resolution is required – to track a single Cdc42 molecule and measure its GTPase cycling. We agree that such data would indeed be interesting, but are unaware of established techniques that would facilitate measurements of Cdc42 cycling rates in vivo.

(2) The reviewer requests in vivo data showing the spatiotemporal interaction of GEF-GAP. Cdc24 and Rga2 are shown to interact (direct or mediated by another protein) (McCusker et al. 2007, Breitzkreutz et al. 2010, Chollet et al. 2020). Cdc24 and Rga2 share 11 binding partners (<https://thebiogrid.org/31724/table/saccharomyces-cerevisiae-s288c/cdc24.html>, <https://thebiogrid.org/32438/table/saccharomyces-cerevisiae-s288c/rga2.html>) and have been found at the polarity spot (Gao et al. 2011). Live cell imaging of fluorescently tagged Cdc24 and Rga2 will show that they exhibit some interaction, but not specify the role of the interaction nor if the interaction is direct or mediated by one of the shared binding partners. In order to show a direct interaction between Cdc24 and Rga2, one could consider (A) super-resolution imaging or (B) FRET experiments: For both fluorescently tagged Cdc24 and Rga2 cell lines would need to be constructed.

(A) Super-resolution imaging could show direct interaction between Cdc24 and Rga2, but even with the techniques available this would be on the limit. Further, it is usually done in fixed cells, and not in live cells (as requested from the reviewer).

(B) To show a direct interaction of Cdc24 and Rga2 using FRET, suitable protein constructs would need to be engineered. We believe that the main obstacle in showing direct binding of Cdc24 and Rga2 using FRET is to design the fluorophore linker. The linker would need to be designed in such a way that it is flexible enough to give a FRET signal even if the two large proteins bind on the opposite sites of the fluorophore, but also is stiff/short enough to *not* show binding if both proteins are in close proximity through binding to a common binding partner (Fig. 1 in this rebuttal).

We believe that an investigation of GEF GAP binding in vivo is beyond the scope of this study. Instead, we explored one possible mechanism underlying GEF GAP synergy - Cdc24 Rga2 binding - through in vitro experiments, finding weak binding between Cdc24 and Rga2 (Fig. 6).

Figure 1 Potential pitfalls in showing direct Cdc24 – Rga2 binding using FRET: (A) No FRET even though proteins bind directly. (B) FRET even though proteins do not bind directly.

Reviewer #1 (Significance (Required)):

The revised study would provide first line evidence that GEF-GAP synergy to be general regulatory property in eukaryotic kingdom.

Reviewer #2 (Evidence, reproducibility and clarity (Required)):

The study entitled, "The GEF Cdc24 and GAP Rga2 synergistically regulate Cdc42 GTPase cycling" by Tschirpke et al., uses an in vitro GTPase assay to examine the GTPase cycle of Cdc42 in combination with its GEF and GAP effectors. The authors find that the Cdc24 GEF activity scales non-linearly with its concentration and the GAP Rga2 has substantially weaker effect on stimulating Cdc42 GTPase activity. Not surprisingly, the combined addition of Cdc24 and Rga2 lead to a substantial increase in Cdc42 GTPase activity.

****Referees cross-commenting****

In Zheng, Y., Cerione, R., and Bender, A. (1994) J. Biol. Chem. 269: 2369-2372 (Fig. 3C), the authors show that Cdc24 combined with the GAP Bem3 lead to a large synergy in boosting Cdc42 GTPase activity.

Reviewer #2 (Significance (Required)):

There is very little new information in this manuscript. Previous studies (Rapali et al. 2017) have shown that the scaffold protein Bem1 enhances the GEF activity of Cdc24. It is expected that the reconstitution of a GEF and GAP protein promote the GTPase cycle and indeed Zheng et al. (1994) showed that that Cdc24 combined with the GAP Bem3 lead to a large synergy in boosting Cdc42 GTPase activity. Hence the only potentially interesting finding in this work is that, in solution Cdc24 activity scales non-linearly with its concentration. However as this GEF and Cdc42 are associated with the membrane, the relevance of solution studies are less clear and furthermore the mechanistic basis for the non-linearity is not explored in detail. Given the limited new information from this work, the findings are, in their current form, too preliminary.

Response from the authors:

We thank the reviewer for their comments. Due to their feedback, we extended the scope of the manuscript in several key aspects:

- We strengthened the link between the observed non-linearity and Cdc24 oligomerization through adding GTPase assay data on an oligomerization-deficient Cdc24 mutant (Fig. 3c and S2).
- We strengthen our hypothesis that the saturation effect of Rga2 is due to self-inhibition, as additional experimental data on Rga2's GAP domain shows that the effect of the GAP domain does not saturate (Fig. 4c).
- We confirm that the Rga2-Cdc24 synergy is protein specific through conducting GTPase assays with Cdc24 and Rga2's GAP domain (which do not exhibit synergy, Fig. 5b).
- We show that Rga2 forms oligomers (Fig. 4a) and that Rga2 and Cdc24 exhibit weak binding (Fig. 6) (new findings).

The reviewer states that the GEF GAP synergy is to be expected, as it was already shown in Zheng et al. 1994. In Fig. 3C Zheng et al. shows the time course of the GTPase activity of Cdc42 in presence of Cdc24, Bem3, and Cdc24 plus Bem3. Fig. 3C is the only data in which the combined effect of a GEF (Cdc24) and a GAP (Bem3) is investigated. The data indicates synergy, but is neither discussed as such in the text of the publication, nor analyzed quantitatively. Zheng et al. 1994 gives an early indication of GEF GAP synergy, but does not claim, discuss, or further investigate the synergy as such. We thank the reviewer for pointing out the pioneering character of Zheng et al.'s study. However, we disagree that Zheng et al. sufficiently studied the GEF GAP interaction. Further, **we added GTPase assay data with Rga2's GAP domain and Cdc24, which together do not exhibit synergy (Fig. 5b). This strongly suggests that the synergy between Rga2 and Cdc24 is protein-specific.**

The reviewer criticizes the relevance of bulk *in vitro* studies (lacking membranes) of proteins that bind to membranes *in vivo*. We agree that the presence of a membrane can affect the protein's property, and we can not exclude that membrane-binding could alter the magnitude of a GEF GAP synergy. However, **we believe that membrane-binding does not impede the GEF GAP synergy altogether.** If membrane binding would influence GTPase properties that strongly, other studies on Cdc42's GTPase activity and GEF and GAP activity, that do not include a membrane, would be inconclusive as well (e.g. Zheng et al. 1993, Zheng et al. 1994, Zheng et al. 1995, Zhang et al. 1997, Zhang et al. 1998, Zhang et al. 1999, Zhang et al. 2000, Zhang et al. 2001, Smith et al. 2002, Rapali et al. 2017). Both studies mentioned by the reviewer (Zheng et al. 1994, Rapali et al. 2017) were also conducted without membranes present.

We believe that an inclusion of membrane-binding into reconstituted Cdc42 systems will enhance our understanding of Cdc42 and recognize it as a next step, which could be enabled by the assay used in our study.

Reviewer #3 (Evidence, reproducibility and clarity (Required)):

This work reports a biochemical analysis of the effects of a recombinant yeast GEF (Cdc24) and GAP (Rga2) on Cdc42 GTPase cycling in vitro. The central conclusion is that the GEF and GAP act "synergistically", which occurs "due to proteins enhancing each other's effects". By this they appear to mean that the GEF enhances the GAP's activity and vice versa. I was not persuaded that this is correct, and was confused by many aspects of the approach and interpretation, as outlined below.

1. GEF and GAP are expected to accelerate GTPase cycle synergistically even with no effect on each other's activity:

The Cdc42 GTPase cycle is understood to occur via distinct steps (GDP release, GTP binding, and GTP hydrolysis): GDP release and GTP hydrolysis are intrinsically slow steps that are accelerated by GEFs (GDP release) and GAPs (GTP hydrolysis). This fundamental biochemistry was established in the 1990s using biochemical assays that measure each step independently. Here instead the authors use an assay that measures [GTP] decline in a mix with 5 uM starting GTP, 1 uM Cdc42, plus or minus some amount of GEF or GAP. They assume exponential decline of [GTP] with time, yielding a cycling "rate". If that is so, then one would expect that added GEF would accelerate only the first step, leaving a slow GTP hydrolysis step that limits the overall cycling rate, while added GAP would accelerate only the last step, leaving a slow GDP release step that limits the overall cycling rate. Adding both together would speed up both steps, and should therefore "synergistically" accelerate cycling. This would be expected based on previous work and does not imply that GEF or GAP are affecting each other's action (except trivially by providing substrate for the next reaction). If the authors wish to demonstrate that something more complex is indeed happening, they need to use assays that directly measure the sub-reaction of interest, as done by prior investigators.

Response from the authors:

The reviewer raises the point that we do not consider a simpler, rate-limiting model and that this rate-limiting model could explain our synergy between GAP and GEF in accelerating the GTPase cycle. We very much welcome this consideration of the reviewer! We added a clarification to our manuscript to explain why a rate-limiting model/interpretation does not match our data (main text and S5). Further, **we added GTPase assays with Rga2's GAP domain and Cdc24, which together do not exhibit synergy (Fig. 5b)**. This strongly suggests that the synergy between Rga2 and Cdc24 is protein-specific.

2. Model-based interpretation of the GTPase assay is poorly supported:

The assay employed measures overall GTP concentration with time. It is assumed (but not well documented-see below) that [GTP] declines exponentially, and that the rate constant for a particular condition can be fit by the sum of a series of terms that are linear or quadratic in the concentrations of Cdc42, GEF, and GAP. There is no theoretical derivation of this model from the elementary reactions, and the assumptions involved are not well articulated.

As discussed in point 1 above, one would expect that a GEF or GAP alone could only accelerate the cycle to a certain point, where the other (slow) reaction becomes rate limiting. But that does not appear to be true for their phenomenological model, where slow steps (small terms in the sum) will always be overwhelmed by fast steps. This is not the traditional understanding of how GTPases operate.

Response from the authors:

The reviewer expresses the concern that because we do not derive our coarse-grained model from elementary reactions, we miss important effects that can occur when adding GAP and GEFs, particularly saturation.

We understand the concern of the reviewer that if a rate-limiting step model is considered, saturation effects of GAP/GEF will limit the amount with which these effectors can speed up the total cycle. Our coarse-grained model indeed does not account for this saturation. To address this concern, we added (1) a mathematical discussion showing that the rate-limiting step model does not explain our data and (2) GTPase assays with Rga2's GAP domain and Cdc24, which together does not exhibit synergy. Secondly, we agree that for high enough concentrations of GEF and GAPs, we would experience a saturation in the effect of adding the effectors. We are aware of this possibility, and we verify that we are not in saturation regimes with our added proteins by checking the plots of the individual protein titrations (see Figure 3, 4). If we enter the saturation regime, we expect a negative second derivative in the rate as function of protein concentration (the curve shallows off). We do not see this for any protein except for Rga2 at some point, as discussed in our main text of the manuscript. However, for this protein we only use the data in the linear regime for further analysis. In short, **we understand the concern of the reviewer but we empirically check that we are not in the saturation regime.**

3. Data that do not conform to expectation are not explained:

Strangely, the data (as interpreted by the model assumptions) also appear inconsistent with the expectation of rate-limiting steps. GEF addition (alone) is said to accelerate cycling 100-fold, while GAP addition (alone) accelerates it 2-fold. But that would seem to imply that GDP release takes up >99% of the basal cycle (so accelerating that step alone reduces cycling time 100-fold), while GTP hydrolysis takes up >50% of the basal cycle (so accelerating that step alone reduces cycling time 2-fold). In the conventional understanding of GTPase cycles, these cannot both be true (as the steps would then add to >100% of the basal cycle). There is no attempt to reconcile these findings with previous work.

Response from the authors:

The reviewer raises the point that our findings do not match the expectations of the rate-limiting model perspective.

We fully agree with the reviewer that our data is not compatible with the rate-limiting step model (discussed in S5). Also, our lack of saturation as described in the previous point of the reviewer provides another argument against using expectations based on rate-limiting steps to interpret our findings.

4. Lack of detailed timecourse data:

The decline in [GTP] with time is stated to be exponential, allowing extraction of an overall cycling "rate". But this claim is supported only weakly (S3 Fig. 1 uses only 3 timepoints, is not plotted on semi-log axis, and does not report fit to exponential vs other models) and only for the Cdc42-alone scenario: no data at all are presented to support exponential decline in reactions with GEF or GAP. Most assays seem to measure only a single timepoint, so extraction of a "rate" is very heavily influenced by the unsupported assumption of exponential decline. And if the decline is not exponential, it becomes extremely difficult to interpret what a single timepoint means.

Response from the authors:

The reviewer requests additional timeseries data with GEF and GAP to support the assumption of an exponential decline of GTP in the assay and requests to plot it on a semi-log axis.

We added data for Cdc42 + Cdc24 and for Cdc42 + Rga2 with two to three time points, and plot it as requested on a semi-log axis (S1 Fig.1).

5. Other issues with interpretation of the data:

(i) It is unclear why the authors chose to employ an assay that is much harder to interpret than the biochemical assays used by others. In biochemical studies, assays that report an output of multiple reactions are always harder to interpret than assays targeting a single reaction. As well-established assays are available for each individual step in GTPase cycles, any conclusions must be supported using such assays.

Response from the authors:

The reviewer wonders why an assay that investigates several GTPase steps at once was chosen over assays that investigate sub-steps of the GTPase cycle, given that these give more mechanistic insights. We agree that assays investigating GTPase cycle sub-steps can give more mechanistic insights into these specific steps. However, they do not allow to study how proteins affecting different steps act together. We were interested in investigating the overall GTPase cycle of Cdc42 and a possible interplay of GEFs and GAPs. Cdc42 GTPase cycling was found to be a requirement for polarity establishment (Wedlich-Soldner et al. 2004) and Cdc42 GTPase cycling is physiologically relevant. Ultimately, we hope that in vitro results provide stepping stones towards understanding the complex and less controlled in vivo environment. The in vivo environment often entails the output of many reactions combined, so there is every incentive to study aggregated effects of a full cycle which are not necessarily the sum of individual outputs.

We believe that both assay types – assays that investigate sub-steps and yield mechanistic details, and assays that investigate the entire cycle – are important and disagree that one assay type is superior to the other. Instead, we believe they complement each other.

(ii) The reported basal (and GEF/GAP-accelerated) rates are very slow, perhaps due to poor folding of recombinant proteins. This raises the possibility that much of the Cdc42 is inactive. If so, then accelerated GTP hydrolysis could come from increasing the active fraction of Cdc42, rather than catalyzing a specific step.

Response from the authors:

The reviewer wonders whether the reported rates are slow due to poor folding of recombinant Cdc42. We used *S. cerevisiae* Cdc42, for which it has been shown that it has a significantly lower basal GTPase activity than Cdc42 of other organisms (see Zhang et al. 1999). Many other studies on Cdc42 were conducted with human Cdc42, which has a significantly higher basal GTPase activity (Zhang et al. 1999). We assessed the activity of several recombinantly expressed Cdc42 constructs previously (Tschirpke et al. 2023). We there observed that most constructs had a similar GTPase activity, only some purification batches and constructs had a significantly reduced GTPase activity (which might be linked to poor folding). **The Cdc42 construct used here shows a similar activity as the active Cdc42 constructs in Tschirpke et al. 2023, and we therefore believe that it exhibits proper folding.** If recombinant Cdc42 folds poorly, we would expect greater variations between Cdc42 constructs and purification batches (caused by different levels of folding/ a different fraction of active Cdc42) than what we observed previously (see Tschirpke et al. 2023).

Tschirpke et al. 2023: Tschirpke et al. A guide to the in vitro reconstitution of Cdc42 activity and its regulation (2023) BioRxiv. (<https://doi.org/10.1101/2023.04.24.538075>)

(iii) The GEF and GAP preparations include multiple partial degradation products and it is unclear whether the measured activities come from full-length proteins or more active fragments.

Response from the authors:

We agree with the reviewer that the Cdc24 and Rga2 preparations contain degradation products.

It would be more ideal if the protein purifications were entirely pure, but this is experimentally very difficult to achieve for the used proteins (which are large and partially unstructured, making them prone to partial degradation). Further, it is not uncommon to use protein preparations where some degradation products were present (e.g. Zheng et al. 1993, Zheng et al. 1994). Other studies did not show their purified preparations.

The vast majority of the Cdc24 preparation is the full-length protein. We therefore expect that the degradation fragments only contribute in a small extend to the overall protein behavior.

The Rga2 preparation contains a higher amount of degradation product, but only larger size protein fragments (> 60kDa), suggesting that the fragments contain at least and more than 1/3 of the full-length protein (the protein fragments are thus the size or larger than of the GAP peptides used previously). **We conducted additional GTPase assays using only Rga2's GAP domain (Fig.4c), which showed a distinct concentration-dependent behavior (that is different from Rga2), suggesting that the effect of Rga2 does not originate from GAP domain fragments.**

(iv) Cdc42 cycling is also accelerated by BSA and casein, suggesting that there are poorly understood aspects of the assay and that GEF and GAP actions may (like BSA and casein) involve non-canonical effects on Cdc42. As GEF and GAP are expected to interact better with Cdc42 than BSA or casein, these effects could dominate the observed changes in GTP levels.

Response from the authors:

The reviewer raises the concern that the effects of the added effector proteins on the rates could be caused by non-canonical effects. **We do not believe non-canonical effects play a relevant role in our assays.** While BSA and casein accelerate the GTPase cycle in our assays, the GAP effect and GEF effect are several times stronger (S4).

(v) Cdc42-alone cycling assays are said to be reproducible. However, assays with added GEF/GAP/BSA/Casein yield rates that vary almost an order of magnitude between replicates. This poor reproducibility further reduces confidence in the findings.

Response from the authors:

The reviewer is concerned about the variations in Cdc42 effector rates.

We disagree that the variations are concerning and believe to have accounted for them in our analysis: The Cdc42 (Cdc42 alone) data is very reproducible (see Tschirpke et al. 2023). The GTPase assay is generally sensitive to small concentration changes and errors introduced through pipetting small volumes (as required for the assay). We believe that the small variation observed for Cdc42 alone is because Cdc42 has such a low basal rate and therefore the small concentration changes due to pipetting have a smaller effect. Once other effectors are added, especially highly GTPase stimulating ones as Cdc24, small concentration changes due to pipetting can lead to larger variations between assays (small variations in Cdc24 concentration lead to larger changes in remaining GTP due to Cdc24's strong and non-linear effect on Cdc42). We conduct the assays multiple times to account for these variations. In our analysis we do not compare single rate numbers but the orders of magnitude of the rate, and report the variations present. Even given the present variations, the differences in effect sizes are still significant. We map and discuss assay variation in (Tschirpke et al. 2023), to which we refer to several times throughout the manuscript.

Tschirpke et al. 2023: Tschirpke et al. A guide to the in vitro reconstitution of Cdc42 activity and its regulation (2023) BioRxiv. (<https://doi.org/10.1101/2023.04.24.538075>)

(vi) It is unclear what timepoint was used for the different assays. 1.5 h at 30 degrees seems to be the standard here for the Cdc42-alone assays, but I assume that cannot be what was measured to assess GTP decline for GEF-containing assays as there would be very little GTP left at 1.5 h.

Response from the authors:

We used 60-100 min as incubation times for *all* assays. The assay data will be published on a data server, where all these numbers can be checked. **We added a clarification to the materials and methods section** (GTPase assays: 'The reaction mixture got incubated for 60 to 100 min'). In order to still have remaining GTP for the Cdc42 GEF mixtures after 60-100 min, we lowered the used protein concentrations.

(vii) The graph reporting GEF activity is plotted only for $[GEF] < 0.2 \mu M$, but the rates used in the subsequent experiments are reported for mixtures with 1 μM GEF. The full range of GEF data should be plotted.

Response from the authors:

The graphs show the full range of protein concentrations used. In order to calculate K_1 , K_2 , $K_{3,Cdc24}$, $K_{3,Rga2}$, $K_{3,Cdc24,Rga2}$ from k_1 , k_2 , $k_{3,Cdc24}$, $k_{3,Rga2}$, $k_{3,Cdc24,Rga2}$, ..., a protein concentration has to be included in the term (as $K_1 = k_1 [Cdc42]$, ...). In order to make K comparable, we *chose* to use 1 μM for all protein concentrations. This was done to compare the cycling rate values of different proteins. 1 μM was a choice, in the same fashion 0.2 μM could have been chosen. **We added a concentration-dependent plot of the rates (Fig. 5c).**

(viii) S8 Data with casein seems very noisy and it is no longer at all clear that the quadratic fit for $[Cdc24]$ is justified. Also, the symbol colors are very similar so it is hard to tell what data corresponds to what condition. The synergy between Cdc24 and Rga2 is also very noisy and the fits seem arbitrary.

Response from the authors:

The reviewer is concerned with (1) the noise in the S8 data, and (2) the Cdc42-Cdc24-Rga2 fits.

(1) We acknowledge in the manuscript that the S8 data is noisy and would require more replicates. **We removed the data from the publication.**

(2) We disagree that the Cdc42-Cdc24-Rga2 fits are arbitrary. The fits contain several data points per protein, and reproduce the rate values from Cdc42-Cdc24 and Cdc42-Rga2 assays well.

The reviewer is concerned with the color scheme choice in the fits. **We adapted the color scheme of the fits to make the colors more distinguishable.**

(ix) It is disturbing that different Cdc42 constructs behave quite differently (S4). This suggests that protein behavior is influenced by the various added epitope tags and protease cleavage sites (they also leave the C-terminal CAAX box rather than removing the AAX as would happen in vivo). These features raise the concern that these findings may not be directly relevant to the situation with endogenous yeast Cdc42. Of course, it is also the case that relevant Cdc42 biochemistry occurs with prenylated Cdc42 on membranes.

Response from the authors:

The reviewer is concerned that the behavior of the Cdc42 constructs is influenced by their tags. In a previous manuscript (Tschirpke et al. 2023) we explored the effect of various N- and C-terminal tags

on Cdc42, by comparing it to Cdc42 that is not tagged in that position. **We found that most tags, including the tags present in the Cdc42 construct used here, do not affect Cdc42's properties. Instead, we found a general, tag independent, heterogeneity in Cdc42 behavior** (which can occur between purification batches and between constructs (but not between different assays)): in some batches GTPase activity depended quadratically on its concentration, others showed a linear relationship. Most batches exhibited a mixed behavior. The differences between the batches are generally small, and only visible in the activity to concentration plots and because of the assay's high accuracy. We use a two-parameter fit ($k_1 [Cdc42] + k_2 [Cdc42]^2$) to phenomenologically account for this heterogeneity, and to estimate the basal Cdc42 GTPase activity. We do not interpret this heterogeneity, as more research is needed. We believe that Cdc42 still has unexplored properties, of which this heterogeneous behavior can be one. We speculate in Tschirpke et al. 2023 that it is linked to Cdc42 dimerization mediated by its polybasic region, a relationship that is far from being fully understood yet. **We believe that it is of scientific interest to point out heterogeneous behaviors to encourage more research.**

Tschirpke et al. 2023: Tschirpke et al. A guide to the in vitro reconstitution of Cdc42 activity and its regulation (2023) BioRxiv. (<https://doi.org/10.1101/2023.04.24.538075>)

The reviewer is concerned that our findings are biologically not relevant, as our experiments (1) included Cdc42 that was not prenylated and (2) did not include membranes.

(1) We here used recombinantly purified proteins, which do not contain posttranslational modifications, such as prenylations. So-far Cdc42's prenyl group, which is responsible for binding it to membranes, has not been linked to its GTPase properties. We therefore believe that unprenylated Cdc42 is an equal choice to prenylated Cdc42 when studying Cdc42's GTPase cycle. Further, the use of recombinantly purified proteins can be of advantage: when proteins are purified from their native host, the post-translationally modified protein is purified. However, many proteins contain a multitude of post-translational modifications (PTMs). Thus, the purified protein is a mixture of protein with different PTMs. For example, *S. cerevisiae* Cdc42 undergoes ubiquitinylation (Swaney et al. 2013, Back, Gorman, Vogel, & Silva 2019), phosphorylation (Lanz et al. 2021), farnesylation and geranylgeranylation (Caplin, Hettich, & Marshall 1994). We here used protein preparations that do not contain PTMs, and show how they behave. Natively purified proteins would be mixtures of various PTMs, and the observed protein behavior would be that of the mixture. If Cdc42's PTMs affect its GTPase behavior, the observed behavior of natively purified Cdc42 would represent the average behavior of the mixture. It then would require additional work to disentangle which PTMs affect the GTPase cycling in which way. The use of recombinantly expressed Cdc42 does not require this work, and can set the baseline for how Cdc42 without PTMs behaves. If in the future a link between Cdc42's GTPase behavior and PTMs are found, the work here could be used as a baseline for Cdc42's behavior when it is without PTMs.

(2) The concern about missing membranes was also raised by reviewer 2 (significance), and we like to refer to our response there.

Reviewer #3 (Significance (Required)):

The basic biochemistry of Cdc42 cycles was figured out about 30 years ago. However, those studies did not examine how combinations of Cdc42 regulators (as opposed to individual regulators) might interact to produce effects not expected from combining their individual actions. Recently, this combination approach did lead to interesting findings by Rapali et al. This approach is worthwhile and addresses a major question of interest to the broader field of GTPase biochemistry.

One main limitation of this study is technical: the main assay is less informative (though perhaps easier) than traditional assays, and it is unclear whether the recombinant proteins employed retain

their normal activities. Another limitation is the model-based interpretation of the assay that does not include the potential for rate-limiting steps.

Response from the authors:

We thank the reviewer for their comments. Due to their feedback, we extended the scope of the manuscript in several key aspects:

- We strengthened the link between the observed non-linearity and Cdc24 oligomerization through adding GTPase assay data on an oligomerization-deficient Cdc24 mutant (Fig. 3c and S2).
- We strengthen our hypothesis that the saturation effect of Rga2 is due to self-inhibition, as additional experimental data on Rga2's GAP domain shows that the effect of the GAP domain does not saturate (Fig. 4c).
- We confirm that the Rga2-Cdc24 synergy is protein specific through conducting GTPase assays with Cdc24 and Rga2's GAP domain (which do not exhibit synergy, Fig. 5b).
- We show that Rga2 forms oligomers (Fig. 4a) and that Rga2 and Cdc24 exhibit weak binding (Fig. 6) (new findings).

The reviewer finds our assay, which investigates the GTPase cycle as a whole, less informative. Assays investigating single GTPase cycle sub-steps give more mechanistic insights into these steps. We opted for an assay that studies GTPase cycling as a whole instead, as we were interested in studying how proteins effecting different steps act together. We believe that both assay types are important as they complement each other.

The reviewer is concerned about our use of recombinant proteins, and whether they retain their normal activities. We assessed Cdc42's GTPase activity and the influence of added purification tags extensively (Tschirpke et al. 2023), and found that added tags do not affect Cdc42's GTPase properties. We checked Cdc24's GEF activity using the GTPase assay and found that it bound strongly to Bem1, as expected (Tschirpke et al. 2023). The Cdc24 concentrations needed to affect Cdc42's GTPase activity were similar to those used previously (Rapali et al. 2017), suggesting that it is fully active. A similar comparison for Rga2 was not possible, as so-far only domains of Rga2 were used (Smith et al. 2002). We here used recombinantly purified proteins, which do not contain posttranslational modifications (PTMs). To our knowledge the PTMs of the herein used proteins are not linked to their GTPase/GEF/GAP properties. Thus, a lack of PTMs does not diminish our findings. Further, when proteins are purified from their native host, the post-translationally modified protein is purified. However, many proteins contain a multitude of post-translational modifications in vivo. Natively purified proteins would be mixtures of various PTMs, and the observed protein behavior would be that of the mixture. We here used protein preparations that do not contain PTMs, and show how they behave, setting the baseline for proteins without PTMs behaves. If in the future a link between GTPase behavior and PTMs are found, the work here could be used as a baseline for the proteins behavior when it is without PTMs.

Reviewer #4 (Evidence, reproducibility and clarity (Required)):

Summary

The GTPase *cdc42* is a key determinant of yeast polarization. Its activity is amplified at the site of polarization through a poorly defined positive feedback mechanism, and depends on numerous GAPs regulating GTP hydrolysis and the GEF *cdc24* that regulates GDP release. These components have previously been evaluated for their quantitative effects on the individual steps in the GTPase cycle that they modulate, but potential interactions between the *cdc24* GEF and any GAP could not be examined based on these assays. The authors validate and employ a bulk assay of the total GTPase cycle based on GTP consumption to study the activities of and potential interactions between *cdc24* and the GAP *Rga2*. Fitting their data to a mathematical model, they come to three central conclusions: (1) the activating activity of *cdc24* to activate *cdc42* GTPase activity is nonlinear, showing a quadratic relationship, (2) *Rga2* shows a much lower activating activity that is linear at low levels before saturating, and (3) there is a strongly synergistic interaction between the activating activities of *cdc24* and *Rga2*. Some hypotheses for the mechanistic bases of these findings are hypothesized, but not further investigated. Their conclusions are well supported by the data which appears to be of sufficient rigor.

Major comments

The three main conclusions of the manuscript are well supported by the data and associated modeling.

One unresolved issue is the discrepancy between the authors' conclusion that the non-linear activation by *cdc24* is likely a result of oligomerization, whereas Mionnet et al 2008 reach the opposite conclusion. It seems that the authors wish to discount the Mionnet results because they used truncated constructs to test deficient oligomerization and an engineered construct to test induced oligomerization. If the authors are correct, then a relatively easy test would be to introduce the oligomerization deficient mutants defined by Mionnet into their full length construct and compare to wild type protein. While the authors' measured results don't depend on the offered mechanism and this experiment is therefore optional, their explanation is quite unsatisfying, especially since an experiment to resolve the difference is entirely feasible and not very strenuous.

Response from the authors:

The reviewer suggests to conduct experiments with oligomerization deficient *Cdc24* mutants to test our hypothesis that the non-linear concentration dependence of *Cdc24*'s activity is due to *Cdc24* oligomerization. **We conducted such experiments, confirming our hypothesis (Fig. 3c and S2).**

Minor comments

The results in Fig S4 serve as assay validation, and this should be pointed out early in the Results section. I was initially concerned when the assay was described as based on consumption of GTP that a significantly diminished pool would alter the rate and thereby distort results, and being made aware of the S4 result would have alleviated that concern as I read further.

Response from the authors:

We appreciate this suggestion and now mention it earlier (Results – *Cdc42* GTPase activity can be reconstituted in vitro: 'The decrease of the GTP concentration during GTPase cycling is well fitted by an exponential model (S1 Fig. 1)').

On page 4 and Fig S4 the authors mention several cdc42 constructs, some of which show linear activity curves and others slightly non-linear curves. I was unable to find where these constructs or their differences are discussed. The authors should also tell us if the construct used for the remaining experiments was one of the two shown in S4, or a different one.

Response from the authors:

We removed this supplement, as it is extensively discussed in two other publications to which we refer to instead:

- [Tschirpke et al., 2024] Tschirpke, S., Daalman, W. K., and Laan, L. (2024). Quantification of GTPase Cycling Rates of GTPases and GTPase:Effector Mixtures Using GTPase Glo Assays. *Current Protocols*, 4(4):1–29.
- [Tschirpke et al., 2023] Tschirpke, S., van Opstal, F., van der Valk, R., Daalman, W. K.-G., and Laan, L. (2023). A guide to the in vitro reconstitution of Cdc42 activity and its regulation. *BioRxiv*.

It seems that in Fig 4 and Fig S8, some points are missing from the graphs. Were all concentrations for each condition not always assayed, or is some data omitted for some reason? For example, for the 0.125 microM Rga2 condition, only two points are shown vs 4 for some other conditions, and the two missing ones are expected to not be excluded by the >5% GTP remaining criterion.

Response from the authors:

The reviewer wonders whether Fig.4 (now: Fig. 5) and Fig. S8 (now: S4) miss data points. This is not the case, and **we added clarifying information to the manuscript** (materials and methods – GTPase assay: ‘For feasibility the three-protein assays include a reduced number of effector concentrations (in comparison to the two-protein assays)’).

In these graphs, a diamond symbol of slightly varying color is used for the different conditions. The different colors are hard to distinguish. Please use different shape symbols for the different conditions, and choose colors that are more distinct.

Response from the authors:

We adapted the color scheme of the fits to make the colors more distinguishable.

There are a few sentences that are of unclear meaning, for example on page 10, "It was suggested that each GAP plays a distinct role in Cdc42 regulation, of which the level of GAP activity could be a part of [Smith et al., 2002]." There are also typos and grammatical errors that should be fixed.

Response from the authors:

We checked the document for potentially unclear sentences and clarified them, and checked for grammatical and spelling errors.

Reviewer #4 (Significance (Required)):

Significance

The most novel and important finding is the strong synergy observed between cdc24 and Rga2 in activating cdc42 GTPase activity. This is undoubtedly an important mechanism underlying positive

feedback in polarization. The measured non-linear activity of cdc24 alone is also quite important given that availability of cdc24 is thought to be a critical in vivo stimulus for polarization. However, the unexplained discrepancy between this result and that of Mionnet leaves one to wonder which result is more reliable. Only Mionnet attempts to directly test whether oligomerization is important in cdc24 activity.

The conclusions are of importance to a broad audience of cell biologists, though the lack of any mechanism for the synergy or the non-linearity of cdc24 activity somewhat diminishes significance.

Note that my expertise and that of my co-reviewer is in the biology, and while we are able to follow the contributions of the modeling, we do not have the expertise to critically evaluate for potential errors or weaknesses in the modeling itself.

Response from the authors:

The reviewer would find mechanistic insights into (1) the non-linear concentration dependence of Cdc24's activity and (2) the Cdc24-Rga2 synergy useful.

(1) We conducted experiments with partially oligomerization deficient Cdc24 mutants, as suggested by the reviewer (Fig. 3c and S2).

(2) We added data on Cdc24-Rga2 binding (Fig. 6) and Rga2 oligomerization (Fig. 4a) that provide further mechanistic details on the origins of Cdc24-Rga2 synergy (Fig. 7).

Dear Dr. Laan

Thank you for the submission of your research manuscript to our journal. I sincerely apologize for the unusual delay in handling your manuscript. Unfortunately, former referee #3, who was most critical about your data and their interpretation, was not available anymore. I have therefore contacted an advisor who is a good expert in the field. The advisor reviewed your response to the referee concerns and considers your argumentation valid in that an interaction between Cdc24 and Rga2 can contribute to a synergistic benefit. Please see the comments from the advisor copied below my signature. The advisor recommended to thoroughly discuss your findings and tone down your conclusions appropriately, stating: "[...] if the authors at least offered a possible explanation(s) as to why they failed to see synergy between Cdc24 and the limit GAP domain of Rga2 in their Discussion, while maintaining their position that the interaction between these two proteins contributed to a synergistic effect, then I would lean toward accepting the work [...]."

I would therefore invite you to revise your study for publication at EMBO Reports. Please follow the suggestions from the advisor and address the concerns in an adequate discussion and in a point-by-point response.

I copy below the formatting requirements at EMBO Reports. Once you have re-submitted your manuscript, we will perform a number of quality control and data checks on it and I will then send you a final decision letter that details requests from the editorial side, that still need attention.

I am also happy to discuss the revision further via e-mail or a video call, if you wish.

*******IMPORTANT NOTE:**

We perform an initial quality control of all revised manuscripts before re-review. Your manuscript will FAIL this control and the handling will be delayed IN CASE the following APPLIES:

- 1) A data availability section providing access to data deposited in public databases is missing. If you have not deposited any data, please add a sentence to the data availability section that explains that.
- 2) Your manuscript contains statistics and error bars based on $n=2$. Please use scatter blots in these cases. No statistics should be calculated if $n=2$.

When submitting your revised manuscript, please carefully review the instructions that follow below. Failure to include requested items will delay the evaluation of your revision.*****

- 1) a .docx formatted version of the manuscript text (including legends for main figures, EV figures and tables). Please make sure that the changes are highlighted to be clearly visible.
- 2) individual production quality figure files as .eps, .tif, .jpg (one file per figure). Please download our Figure Preparation Guidelines (figure preparation pdf) from our Author Guidelines pages <https://www.embopress.org/page/journal/14693178/authorguide> for more info on how to prepare your figures.
- 3) a .docx formatted letter INCLUDING the reviewers' reports and your detailed point-by-point responses to their comments. As part of the EMBO Press transparent editorial process, the point-by-point response is part of the Review Process File (RPF), which will be published alongside your paper.
- 4) a complete author checklist, which you can download from our author guidelines (). Please insert information in the checklist that is also reflected in the manuscript. The completed author checklist will also be part of the RPF.
- 5) Please note that all corresponding authors are required to supply an ORCID ID for their name upon submission of a revised manuscript (). Please find instructions on how to link your ORCID ID to your account in our manuscript tracking system in our Author guidelines ()
- 6) We replaced Supplementary Information with Expanded View (EV) Figures and Tables that are collapsible/expandable online. A maximum of 5 EV Figures can be typeset. EV Figures should be cited as 'Figure EV1, Figure EV2' etc... in the text and their respective legends should be included in the main text after the legends of regular figures.

7) Please note that a Data Availability section at the end of Materials and Methods is now mandatory. In case you have no data that requires deposition in a public database, please state so instead of refereeing to the database.

See also < <https://www.embopress.org/page/journal/14693178/authorguide#dataavailability>>. Please note that the Data Availability Section is restricted to new primary data that are part of this study.

Additional information on source data and instruction on how to label the files are available .

10) Figure legends and data quantification:

- the name of the statistical test used to generate error bars and P values,
- the number (n) of independent experiments (please specify technical or biological replicates) underlying each data point,
- the nature of the bars and error bars (s.d., s.e.m.)

- If the data are obtained from n {less than or equal to} 5, show the individual data points in addition to the SD or SEM.

- If the data are obtained from n {less than or equal to} 2, use scatter blots showing the individual data points.

11) Our journal encourages inclusion of *data citations in the reference list* to directly cite datasets that were re-used and obtained from public databases. Data citations in the article text are distinct from normal bibliographical citations and should directly link to the database records from which the data can be accessed. In the main text, data citations are formatted as follows: "Data ref: Smith et al, 2001" or "Data ref: NCBI Sequence Read Archive PRJNA342805, 2017". In the Reference list, data citations must be labeled with "[DATASET]". A data reference must provide the database name, accession number/identifiers and a resolvable link to the landing page from which the data can be accessed at the end of the reference. Further instructions are available at .

12) All Materials and Methods need to be described in the main text using our 'Structured Methods' format. According to this format, the Methods section includes a Reagents and Tools Table (listing key reagents, experimental models, software and relevant equipment and including their sources and relevant identifiers) followed by a Methods and Protocols section describing the methods, ideally using a step-by-step protocol format. The aim is to facilitate adoption of the methodologies across labs. Please download and fill our Reagents and Tools Table template (.docx), which you can find in our author guidelines: <https://www.embopress.org/page/journal/14693178/authorguide#structuredmethods>.

13) As part of the EMBO publication's Transparent Editorial Process, EMBO Reports publishes online a Review Process File to accompany accepted manuscripts. This File will be published in conjunction with your paper and will include the referee reports, your point-by-point response and all pertinent correspondence relating to the manuscript.

Yours sincerely,

=====

Comments from the advisor:

I have gone through the authors' rebuttal of the various reviewers with special attention to reviewer 3. To be honest, this is a tough judgement call in terms of whether the authors fully satisfied the concerns of reviewer 3. The authors have for the most part made reasonable attempts to respond to the reviewer's queries regarding the source of the synergy between the yeast Cdc42 GEF, Cdc24, and the GAP Rga2. Their most persuasive argument for their contention, that the synergy they observe between Cdc24 and Rga2 is due to an interaction between the two proteins which prevents an auto-inhibition of the Rga2 GAP activity caused by Rga2 oligomerization, is their failure to see synergy when assaying Cdc24 with the limit GAP domain of Rga2. They feel this argues that the synergy is due to interactions between the two regulatory proteins rather than reviewer 3's suggestion that synergy is expected because the two regulators each accelerate slow steps in the GTP-binding/GTP hydrolytic cycle. However, I too would have expected synergy even in the case when using a limit GAP domain, because the GEF, by accelerating GDP dissociation and enabling an increased rate in GTP binding should also give rise to an increased rate of GAP-stimulated GTP hydrolysis (i.e., having both a GEF and a GAP should seemingly always provide a synergistic benefit like reviewer 3 contends).

In the end, I suspect that the authors' might be correct that an interaction between Cdc24 and Rga2 can contribute to a synergistic benefit if that interaction prevents an inhibitory self oligomerization by Rga2. However, I would like to know why the combination of Cdc24 and the Rga2 limit GAP domain failed to provide a synergistic effect (contrary to what reviewer 3 and I would have expected). Does it have to do with their assay conditions or somehow an interaction that also occurs between Cdc24 and the limit GAP domain of Rga2 that alters the GEF and GAP accelerated steps in a way that masks any synergy? I guess I would ask the authors at least to say something in the Discussion of their revised manuscript that tries to address this. If they do, I would think the manuscript could then be accepted.

Advisor:

I have gone through the authors' rebuttal of the various reviewers with special attention to reviewer 3. To be honest, this is a tough judgement call in terms of whether the authors fully satisfied the concerns of reviewer 3. The authors have for the most part made reasonable attempts to respond to the reviewer's queries regarding the source of the synergy between the yeast Cdc42 GEF, Cdc24, and the GAP Rga2. Their most persuasive argument for their contention, that the synergy they observe between Cdc24 and Rga2 is due to an interaction between the two proteins which prevents an auto-inhibition of the Rga2 GAP activity caused by Rga2 oligomerization, is their failure to see synergy when assaying Cdc24 with the limit GAP domain of Rga2. They feel this argues that the synergy is due to interactions between the two regulatory proteins rather than reviewer 3's suggestion that synergy is expected because the two regulators each accelerate slow steps in the GTP-binding/GTP hydrolytic cycle. However, I too would have expected synergy even in the case when using a limit GAP domain, because the GEF, by accelerating GDP dissociation and enabling an increased rate in GTP binding should also give rise to an increased rate of GAP-stimulated GTP hydrolysis (i.e., having both a GEF and a GAP should seemingly always provide a synergistic benefit like reviewer 3 contends).

In the end, I suspect that the authors' might be correct that an interaction between Cdc24 and Rga2 can contribute to a synergistic benefit if that interaction prevents an inhibitory self oligomerization by Rga2. However, I would like to know why the combination of Cdc24 and the Rga2 limit GAP domain failed to provide a synergistic effect (contrary to what reviewer 3 and I would have expected). Does it have to do with their assay conditions or somehow an interaction that also occurs between Cdc24 and the limit GAP domain of Rga2 that alters the GEF and GAP accelerated steps in a way that masks any synergy? I guess I would ask the authors at least to say something in the Discussion of their revised manuscript that tries to address this. If they do, I would think the manuscript could then be accepted.

Rev_Com_number: RC-2023-02060

New_manu_number: EMBOR-2023-58186V2-Q

Corr_author: Laan

Title: Oligomerization-dependent and synergistic regulation of Cdc42 GTPase cycling by a GEF and a GAP

All minor editorial requests have been addressed by the authors.

Dear Liedewij,

Thank you for the submission of your revised manuscript to EMBO reports. I sincerely apologize for the delay in handling your manuscript. I have now checked the manuscript from the editorial side and note a few points below that we will need before we can proceed with the official acceptance of your study.

Could you please also provide a short point-by-point like response to the comments from the advisor or indicate the paragraph in the Discussion where this point has been addressed? Thank you very much.

"[...] if the authors at least offered a possible explanation(s) as to why they failed to see synergy between Cdc24 and the limit GAP domain of Rga2 in their Discussion, while maintaining their position that the interaction between these two proteins contributed to a synergistic effect, then I would lean toward accepting the work [...]."

Minor corrections required from the editorial side:

1. Please provide up to 5 keywords on the title page.

2. The manuscript sections are in the wrong order. Please order them like this:

Title page - Abstract - Introduction - Results - Discussion - Methods - Acknowledgements - Disclosure and competing interests statement - References - Figure legends - Tables and their legends (not EV tables) - Expanded View Figure legends

3. Figure legends and Tables should be provided after the References at the very end of the manuscript.

4. We need the conflict of interest statement in a separate section called "Disclosure and Competing Interests Statement", which is placed after the Acknowledgments.

5. The information on funding needs to be part of Acknowledgments and the separate Funding section heading should be removed.

6. Regarding the Author Contributions, we now use CRediT to specify the contributions of each author in the journal submission system. Therefore, please remove the Author Contributions from the manuscript file and make sure that the author contributions in our online manuscript tracking system are correct and up-to-date. The information you specified in the system will be automatically retrieved and typeset into the article. You can enter additional information in the free text box provided, if you wish.

7. References: et al needs to be used after 10 author names; DOIs should only be used for preprints and datasets that have not been published yet.

8. Citations to preprint articles (e.g., Tschirpke et al 2023) should be labeled as such. In text citations: (preprint: Tschirpke S et al, 2023). In the reference list [PREPRINT] is added at the end of the reference.

9. Please add callouts for Fig. 3d, Fig. 4d, the individual panels of Fig. 7 (a, b), Tables 1-5, and the Supplementary figures.

10. Figure 8a is cited but there is no such panel labeled in the figure.

11. "Supplements" file: This PDF file appears to contain text describing results. Please note that all text/description of results need to be part of the main manuscript file. The Supplement should only contain figures and/or tables, the description of these items should be in the main manuscript text, unless there are good reasons for having them in the Supplement.

12. The nomenclature of supplementary files should be Appendix Figure S1, etc. Appendix Table S1, etc.; The file is called "Appendix" and we would also need a table of content with page numbers.

13. The panel labels in the figure should be capital letters (A instead of a).

14. Please download and fill our Reagents and Tools Table template (.docx), which you can find in our author guidelines:

Tables 4 and 5 can be part of the Reagent and Tools table.

15. At EMBO Press we ask authors to provide source data for the main manuscript figures. You have already received a separate email with instructions for providing source data with your revised manuscript, including how to upload and organize the files.

Additional information on source data and instruction on how to label the files are available

16. The abbreviations section needs to be removed from the manuscript. Abbreviations should be defined in brackets after their first mention in the text, instead of a list of abbreviations.

17. Please note that information related to n is missing in the legends of figures 3A-C; 4B, C; 5A, B

18. Finally, EMBO Reports papers are accompanied online by

A) a short (1-2 sentences) summary of the findings and their significance,

B) 2-3 bullet points highlighting key results and

C) a schematic summary figure that provides a sketch of the major findings (not a data image).

Please provide the summary figure as a separate file in PNG or JPG format at a size of 550x300-600 pixels (width x height).

Please note that the size is rather small and that text needs to be readable at the final size. Please send us this information along with the revised manuscript.

With kind regards,

Martina

Rev_Com_number: RC-2023-02060

New_manu_number: EMBOR-2023-58186V3

Corr_author: Laan

Title: Oligomerization-dependent and synergistic regulation of Cdc42 GTPase cycling by a GEF and a GAP

Response to the Advisor's comment:

Advisor's comment: "[...] if the authors at least offered a possible explanation(s) as to why they failed to see synergy between Cdc24 and the limit GAP domain of Rga2 in their Discussion, while maintaining their position that the interaction between these two proteins contributed to a synergistic effect, then I would lean toward accepting the work [...]"

Response from the Authors: We welcomed the advisor's comment, as it is an important observation we had not explained yet. We added an explanatory figure to the results section (figure 6), and addressed the advisor's point in-detail in the results section "Cdc24 and Rga2 synergistically regulate Cdc42 GTPase cycling" (specific section pasted below). We believe that the advisor's point is better addressed in the earlier results section than in the discussion section.

From the manuscript: "Could rate-limiting steps be the only source of the positive $K_{3,Cdc24,Rga2}$? The rate-limiting step model is a common framework used to describe the cycling behavior of GTPases. It assumes that at least one of the three GTPase cycle steps — GTP binding, GTP hydrolysis, or GDP release — is rate-limiting. It has the potential to explain the synergy term we observed in our data: if both the GDP release step, accelerated by the GEF Cdc24, and the GTP hydrolysis step, accelerated by the GAP Rga2, are rate-limiting, then adding both effectors would accelerate the overall cycle more than either alone. This synergistic increase in cycling speed would result from their combined effect on the GTPase cycle and not from direct interaction between the proteins. The effect size of the synergetic term, as predicted by the rate-limiting step model, depends on the concentrations of the GEF and GAP used: the synergy is strongest when both steps — GDP release and GTP hydrolysis — are rate-limiting, which occurs at higher concentrations of GEF and GAP (Fig. 6b). Conversely, the synergy diminishes when these steps are not rate-limiting, i.e. at lower effector concentrations (Fig. 6a). At which GEF and GAP concentrations do GDP release and GTP hydrolysis become rate limiting? When either step becomes rate-limiting, we expect the overall GTP hydrolysis rate K to saturate (Fig. 6c). We conducted experiments with only one effector present (Cdc42 + Cdc24, Cdc42 + Rga2): we do not observe saturation of K with increasing Cdc24 concentration (Fig. 3b), suggesting that GDP release does not become rate-limiting within our tested concentration range. Although K does saturate with full-length Rga2 (Fig. 4b), it does not with the GAP domain (Fig. 4c), suggesting that we also do not reach concentration regimes where GTP hydrolysis becomes rate-limiting.

Thus, we expect the overall synergy due to rate-limiting steps to be small."

Dr. Liedewij Laan
Delft University of Technology
Bionanoscience Department
Netherlands

Dear Liedewij,

Thank you for implementing the final editorial changes. I am now very pleased to accept your manuscript for publication in the next available issue of EMBO reports. Thank you for your contribution to our journal.

You may qualify for financial assistance for your publication charges - either via a Springer Nature fully open access agreement or an EMBO initiative. Check your eligibility: <https://link.springer.com/journal/44319/how-to-publish-with-us>

Kind regards,

Martina

>>> Please note that it is EMBO Reports policy for the transcript of the editorial process (containing referee reports and your response letter) to be published as an online supplement to each paper. If you do NOT want this, you will need to inform the Editorial Office via email immediately. More information is available here: <https://link.springer.com/partners/embo-press/editorial-policies#Peer%20review>

Rev_Com_number: RC-2023-02060

New_manu_number: EMBOR-2023-58186V4

Corr_author: Laan

Title: Oligomerization-dependent and synergistic regulation of Cdc42 GTPase cycling by a GEF and a GAP